# Systematic Relational Reasoning With Epistemic Graph Neural Networks

**Irtaza Khalid & Steven Schockaert**
Cardiff University, UK
{khalidmi,schockaerts1}@cardiff.ac.uk

## Abstract

Developing models that can learn to reason is a notoriously challenging problem. We focus on reasoning in relational domains, where the use of Graph Neural Networks (GNNs) seems like a natural choice. However, previous work has shown that regular GNNs lack the ability to systematically generalize from training examples on test graphs requiring longer inference chains, which fundamentally limits their reasoning abilities. A common solution relies on neuro-symbolic methods that systematically reason by learning rules, but their scalability is often limited and they tend to make unrealistically strong assumptions, e.g. that the answer can always be inferred from a single relational path. We propose the Epistemic GNN (EpiGNN), a novel parameter-efficient and scalable GNN architecture with an epistemic inductive bias for systematic reasoning. Node embeddings in EpiGNNs are treated as epistemic states, and message passing is implemented accordingly. We show that EpiGNNs achieve state-of-the-art results on link prediction tasks that require systematic reasoning. Furthermore, for inductive knowledge graph completion, EpiGNNs rival the performance of state-of-the-art specialized approaches. Finally, we introduce two new benchmarks that go beyond standard relational reasoning by requiring the aggregation of information from multiple paths. Here, existing neuro-symbolic approaches fail, yet EpiGNNs learn to reason accurately. Code and datasets are available at https://github.com/erg0dic/gnn-sg.

## 1 Introduction

Learning to reason remains a key challenge for neural networks. When standard neural network architectures are trained on reasoning problems, they often perform well on similar problems but fail to generalize to problems with different characteristics than the ones that were seen during training, e.g. problems that were sampled from a different distribution (Zhang et al., 2023a). This behaviour has been observed for different types of reasoning problems and different types of architectures, including pretrained transformers (Zhang et al., 2023a; Welleck et al., 2023), transformer variants (Bergen et al., 2021; Kazemnejad et al., 2023) and Graph Neural Networks (Sinha et al., 2019).

In this paper, we focus on the problem of *systematic generalization* (SG), and on systematic reasoning about binary relations in particular. This refers to the ability of a model to solve test instances by applying knowledge obtained from training instances, where the combination of inference steps that is needed is different from what has been seen during training (Hupkes et al., 2020). It is an essential ingredient for machines and humans to generalize from a limited amount of data (Lake et al., 2017). For relational systematic reasoning, problem instances can be represented as a labelled multi-graph (i.e. a knowledge graph) and the main reasoning task is to systematically infer the relationship between a head entity $h$ and target entity $t$. Typically, the training data consists of instances that only require relatively short inference chains, where trained models are then evaluated on increasingly larger test graphs. Graph Neural Networks (GNNs) intuitively seem well-suited for relational reasoning (Zhu et al., 2021; Schlichtkrull et al., 2018), but in practice they underperform neuro-symbolic methods on SG (Rocktäschel & Riedel, 2017; Minervini et al., 2020b). Crucially, this is true despite the fact that GNNs are expressive enough to encode the required inference process.

Conceptually, there is a fundamental question that has largely remained unanswered: *What makes neural-theorem-prover type methods successful for systematic reasoning?* We argue that the outper-

Figure 1: Left: A (single) relational path reasoning problem over family relations from CLUTRR (Sinha et al., 2019), where the path $P_1$ allows us to infer the correct relation. Right: A multi-path reasoning problem over RCC-8 relations where each path provides partial (disjunctive) information and the target label is obtained by combining information from paths $P_1$, $P_2$ and $P_3$.

formance of such methods can be explained by their focus on modeling *single relational paths*, as an alternative to local message passing. To predict the relationship between $h$ and $t$, conceptually, these methods consider all the (exponentially many) paths between them, select the most informative path, and make a prediction based on this path, although in practice heuristics are used to avoid exhaustive exploration (Minervini et al., 2020b). The emphasis on *single* paths provides a useful inductive bias, while also avoiding problems with local message passing such as over-smoothing. Compared to GNNs, neuro-symbolic methods also have an advantage in how individual relational paths are modeled. Consider a relational path $h \xrightarrow{r_1} x_1 \xrightarrow{r_2} ... \xrightarrow{r_k} t$, or simply $r_1; r_2; ...; r_k$ if the entities $x_i$ are unimportant. To predict the relationship between $h$ and $t$, most methods essentially proceed by repeatedly choosing two neighboring relations $r_i; r_{i+1}$ and replace them by their composition. Crucially, the order in which these relations are chosen may determine whether the model finds the answer, as certain orderings may require knowledge of unseen intermediate relationships. Neuro-symbolic methods can often avoid such issues by, in principle, considering all possible orderings.

However, neuro-symbolic methods also have important drawbacks, including scalability, and most fundamentally, their focus on simple rules and single relational paths. Since these limitations are not tested by existing benchmarks, as a first contribution, we introduce a multi-path, disjunctive systematic reasoning benchmark based on qualitative spatial (RCC-8) and temporal (Interval Algebra) calculi (Randell et al., 1992; Allen, 1983b). The considered problems require models to combine partial (disjunctive) information from multiple relational paths, which are also present in real-world story understanding problems (Cain et al., 2001). Figure 1 illustrates the difference between single and multi-path relational reasoning.

In this paper, we propose the Epistemic GNN (EpiGNN), a novel, scalable and parameter-efficient GNN for systematic relational reasoning. Motivated by the fact that aligning the model's architecture with an algorithm that approximately solves the given problem aids generalization (Xu et al., 2020; Bahdanau et al., 2019), the EpiGNN is designed to simulate an approximation of the Algebraic Closure Algorithm (ACA) (Renz & Ligozat, 2005), which solves multi-path reasoning on RCC-8 and IA. This translates to the following inductive biases in the EpiGNN's structure: (1) having a message passing function that explicitly simulates the composition of discrete relations in ACA (rather than general relation vector composition) (2) having epistemic (probabilistic) embeddings that can encode unions of base relations in ACA (3) having a pooling operation that simulates the intersection operator in ACA. Below is a summary of our main contributions:

- We propose a novel GNN model, the EpiGNN, based on ACA, which rivals SOTA neuro-symbolic methods on simple single-path base systematic reasoning while being highly efficient, and at least two orders of magnitude more parameter-efficient in practice.

- We introduce two multi-path, disjunctive relational reasoning benchmarks that are challenging for various SOTA models, but where EpiGNNs still perform well.

- Despite being designed for SG-type link prediction, we show that EpiGNNs rival SOTA specialized approaches for standard inductive knowledge graph completion.

- We theoretically link EpiGNNs to an approximation of the algebraic closure algorithm, which we term *directional algebraic closure*.

## 2 LEARNING TO REASON IN RELATIONAL DOMAINS

We focus on the problem of reasoning about binary relations. We assume that a set $\mathcal{F}$ of facts is given, referring to a set of relations $\mathcal{R}$ and a set of entities $\mathcal{E}$. Each of these facts is an *atom* of the form $r(a, b)$, with $r \in \mathcal{R}$ and $a, b \in \mathcal{E}$. We furthermore assume that there exists a set of rules $\mathcal{K}$ which can be used to infer relationships between the entities in $\mathcal{E}$. We write $\mathcal{K} \cup \mathcal{F} \models r(a, b)$ to denote that $r(a, b)$ can be inferred from the facts in $\mathcal{F}$ and the rules in $\mathcal{K}$. The problem that we are interested in is to develop a neural network model $f_\theta$ which can predict for a given assertion $r(a, b)$ whether $\mathcal{K} \cup \mathcal{F} \models r(a, b)$ holds or not. Note that the set of rules $\mathcal{K}$ is not given. We instead have access to a number of fact sets $\mathcal{F}_i$, together with examples of atoms $r(a, b)$ which can be inferred from these fact graphs and atoms which cannot. To be successful, $f_\theta$ must essentially learn the rules from $\mathcal{K}$, and the considered model must be capable of applying the learned rules in a systematic way to new problems.

### 2.1 LEARNING TO REASON ABOUT SIMPLE PATH RULES

The most commonly studied setting concerns Horn rules of the following form ($n \geq 3$):
$$r(X_1, X_n) \leftarrow r_1(X_1, X_2) \wedge \ldots \wedge r_{n-1}(X_{n-1}, X_n) \tag{1}$$
We will refer to such rules as *simple path rules*. Note that we used the convention from logic programming to write the head of the rule on the left-hand side, and we use uppercase symbols such as $X_i$ to denote variables. We can naturally associate a labelled multi-graph $G_\mathcal{F}$ with the given set of facts. The rule (1) expresses that when two entities $a$ and $b$ are connected by a relational path $r_1; \ldots; r_{n-1}$ in this graph $G_\mathcal{F}$, then we can infer that $r(a, b)$ is true. Without loss of generality, we can restrict this setting to rules with two atoms in the body: $r(X_1, X_3) \leftarrow r_1(X_1, X_2) \wedge r_2(X_2, X_3)$. Indeed, a rule with more than two atoms in the body can be straightforwardly simulated by introducing fresh relation symbols. The semantics of entailment are defined in the usual way (see Appendix A for details). We can think of the process of showing $\mathcal{K} \cup \mathcal{F} \models r(a, b)$ in terms of operations on relational paths. We say that the path $r_1; \ldots; r_{i-2}; s; r_{i+1}; \ldots; r_k$ can be derived from $r_1; \ldots; r_k$ in one step if $\mathcal{K}$ contains a rule of the form $s(X, Z) \leftarrow r_{i-1}(X, Y) \wedge r_i(Y, Z)$. We say that $r$ can be derived from $r_1; \ldots; r_k$ if there exists a sequence of $k - 1$ such steps that yields $r$.

**Proposition 1.** *We have that $\mathcal{K} \cup \mathcal{F} \models r(a, b)$ holds iff there exists a relational path $r_1; \ldots; r_k$ connecting $a$ and $b$ in the graph $G_\mathcal{F}$ such that $r$ can be derived from $r_1; \ldots; r_k$.*

Inferring $r(a, b)$ thus conceptually consists of two distinct steps: (i) selecting a relational path between $a$ and $b$ and (ii) showing that $r$ can be derived from it. Several of the neural network methods that have been proposed in recent years for relational reasoning directly implement these two steps, most notably R5 (Lu et al., 2022) and NCRL (Cheng et al., 2023). Neural theorem provers (Rocktäschel & Riedel, 2017) implicitly also operate in a similar way, considering all possible paths and all possible derivations for these paths. However, both steps are problematic for standard GNNs. First, by focusing on local message passing, rather than on selecting individual relational paths, the node representations that are learned by a GNN run the risk of becoming "overloaded", as they intuitively aggregate information from all the paths that go through a given node. Moreover, even for graphs that consist of a single path, GNNs have a disadvantage: the use of local message passing intuitively means that relational paths have to be processed sequentially. For example, for $r_1; r_2; r_3$, models such as NBFNet (Zhu et al., 2021) can only process $r_1; r_2$ as the first valid composition and not $r_2; r_3$.

The fact that relational paths can only be processed sequentially by GNNs is an important limitation, as this might require the model to apply rules, or even capture types of relations, that were not present in the training data. For instance, we may encounter the following chain of family relationships:
$$a \xrightarrow{\textit{has-father}} x_1 \xrightarrow{\textit{has-father}} x_2 \xrightarrow{\textit{has-brother}} x_3 \xrightarrow{\textit{has-daughter}} x_4 \xrightarrow{\textit{has-brother}} x_5 \xrightarrow{\textit{has-mother}} b$$
Suppose the model has never encountered the *great-cousin* relation during training. Then we cannot expect it to capture the relational path *has-father*; *has-father*; *has-brother*; *has-daughter*. However, if it can first derive $a \xrightarrow{\textit{has-father}} x_1 \xrightarrow{\textit{has-aunt}} b$ then the problem disappears (assuming the model understands the *great-aunt* relation).

### 2.2 LEARNING TO REASON ABOUT DISJUNCTIVE RULES

The rules that we have considered thus far uniquely determine the relationship between two entities $a$ and $c$, given knowledge about how $a$ relates to $b$ and $b$ relates to $c$. In many settings, however, such

knowledge might not be sufficient for completely characterising the relationship between $a$ and $c$. Domain knowledge might then be expressed using disjunctive rules of the following form:

$$s_1(X, Z) \vee \ldots \vee s_k(X, Z) \leftarrow r_1(X, Y) \wedge r_2(Y, Z) \tag{2}$$

In other words, if we know that $r_1(a, b)$ and $r_2(b, c)$ hold for some entities $a, b, c$, then we can only infer that one of the relations $s_1, \ldots, s_k$ must hold between $a$ and $c$. We then typically also have constraints of the form $\bot \leftarrow r_1(X, Y) \wedge r_2(X, Y)$, encoding that $r_1$ and $r_2$ are disjoint. Many popular calculi for spatial and temporal reasoning fall under this setting, including the Region Connection Calculus (RCC-8) and the Interval Algebra (IA) (Allen, 1983a; Randell et al., 1992). RCC-8 uses eight relations to qualitatively describe spatial relations, as shown in Fig. 2. For instance, $\mathsf{ntpp}(a, b)$ means that $a$ is a proper part of the interior of $b$. IA is defined similarly (see App. E).

Existing benchmarks for systematic relational reasoning do not consider disjunctive rules, and thus only test the reasoning abilities of models to a limited extent. We therefore introduce two new relational reasoning benchmarks, based on RCC-8 and IA respectively. An example RCC-8 problem is shown in Figure 1, illustrating how we may need to aggregate evidence from multiple relational paths, as a single path does not always yield a unique relation. For instance, from path $P_3$ we can only infer that one of the relations $\mathsf{tppi}, \mathsf{ntppi}, \mathsf{po}$ holds between nodes 0 and 10. Full details of the proposed benchmarks are provided in Appendix E. Methods such as R5 and NCRL, which rely on a single path to make predictions, cannot be used in this setting. Neural theorem provers cannot handle disjunctive rules either, and cannot be generalized in a scalable way. We thus need a new approach for for learning to reason about disjunctive rules.

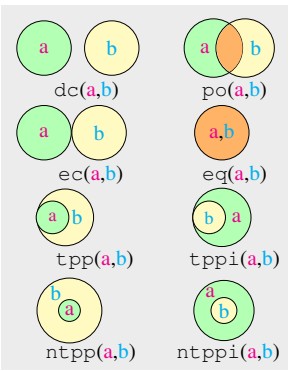

Figure 2: RCC-8 relations.

The characterization in Proposition 1 explains how models can be designed for deciding entailment with simple rules. A similar characterization for entailment with disjunctive rules is unfortunately not possible, as deciding entailment with such rules is NP-complete in general. However, for RCC-8 and IA, and many other calculi, deciding entailment in polynomial time is *possible* using the *algebraic closure* algorithm (Renz & Ligozat, 2005). The main idea is to keep track, for every pair of entities, of which relationships are possible between these entities. This knowledge of possible relationships is then propagated to infer constraints about relationships between other entities using the rules in $\mathcal{K}$ (details are provided in the App. C). As we will see next, our proposed epistemic GNN model is based on the same idea, and can be viewed as an approximate differentiable algebraic closure method.

## 3 AN EPISTEMIC GNN FOR SYSTEMATIC REASONING

We now present the Epistemic GNN (EpiGNN), a GNN which aims to overcome a number of key limitations of existing models for systematic relational reasoning. Specifically, EpiGNNs are more efficient than current neuro-symbolic methods while, as we will see in Section 4, matching their performance on reasoning with simple path rules. Moreover, they are also able to reason about disjunctive rules, which has not been previously considered for neuro-symbolic methods.

We start from the principle that reasoning is fundamentally about manipulating epistemic states (i.e. states of knowledge) and the GNN should reflect this: there should be a clear correspondence between the node embeddings that are learned by the model and what we can infer about the relationships that may hold between the entities of interest. Inspired by NBFNet (Zhu et al., 2021), a GNN that models path-based representations by anchoring node embeddings to a source, we also use a network which learns the relationships between one designated head entity $h$ and all other entities. Let us write $\mathbf{e}^{(l)} \in \mathbb{R}^n$ for the embedding of entity $e$ in layer $l$ of the network. This embedding reflects which relationships may hold between $e$ and the designated entity $h$. We think of $\mathbf{e}^{(l)}$ as a probability distribution over possible relationships. Accordingly, the embeddings are initialized as:

$$\mathbf{e}^{(0)} = \begin{cases} (1, 0, \ldots, 0) & \text{if } e = h \\ (\frac{1}{n}, \ldots, \frac{1}{n}) & \text{otherwise} \end{cases} \tag{3}$$

We associate the first coordinate with the identity relation. Since $h$ is identical to itself, we define $\mathbf{h}^{(0)}$ as $(1, 0, \ldots, 0)$. For the other entities, since we have no knowledge, their embeddings are

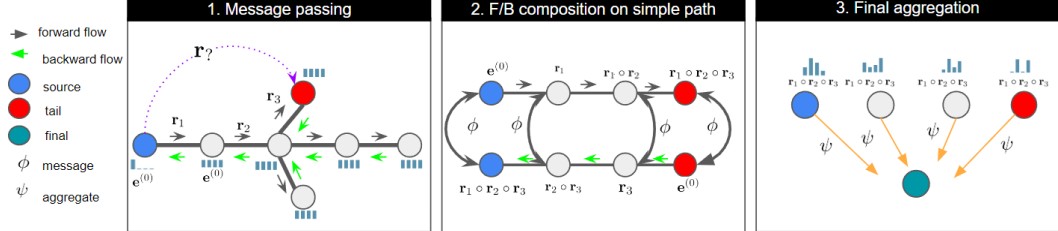

Figure 3: Overview of the EpiGNN. Step 1: Independently learn the forward and backward entity embeddings through epistemic message passing. Step 2: Compose the entity embeddings on a path between the head (blue) and target (red) entity from the forward and backward model. Each composition predicts the target relation. Step 3: Aggregate the evidence provided by each prediction.

initialized as a uniform distribution. Note that the entity components correspond to abstract primitive relations, rather than the relations from $\mathcal{R}$. We only require that the relations from $\mathcal{R}$ can be defined in terms of these primitive relations. This distinction is important, because it allows us to capture semantic dependencies between relations (e.g. both *parent* and *father* may exist in $\mathcal{R}$) and to express (composed) relationships which are outside $\mathcal{R}$. For $l \geq 1$, the embeddings are updated as:

$$\mathbf{e}^{(l)} = \psi(\{\mathbf{e}^{(l-1)}\} \cup \{\phi(\mathbf{f}^{(l-1)}, \mathbf{r}) \mid r(f, e) \in \mathcal{F}\})$$

where the argument of the pooling operator $\psi$ is a multi-set and $\mathbf{r} \in \mathbb{R}^n$ is a learned embedding of relation $r \in \mathcal{R}$, where $\mathbf{r} = (r_1, \ldots, r_n)$ encodes a probability distribution, i.e. $r_i \geq 0$ and $\sum_i r_i = 1$.

**Message passing**  The vector $\phi(\mathbf{f}^{(l-1)}, \mathbf{r})$ should capture the possible relationships between $h$ and $e$, given the knowledge provided by $\mathbf{f}^{(l-1)}$ about the possible relationships between $h$ and $f$ and the fact $r(f, e)$. Since both $\mathbf{f}^{(l-1)}$ and $\mathbf{r}$ are modeled as probability distributions over primitive relations, $\phi(\mathbf{f}^{(l-1)}, \mathbf{r})$ can be defined in terms of compositions of primitive relations. Specifically, let the vector $\mathbf{a}_{ij} \in \mathbb{R}^n$ represent the composition of primitive relations $i$ and $j$, which we treat as a probability distribution, i.e. we require the components of $\mathbf{a}_{ij}$ to be non-negative and to sum to 1. We define:

$$\phi((f_1, \ldots, f_n), (r_1, \ldots, r_n)) = \sum_{i=1}^{n} \sum_{j=1}^{n} f_i r_j \mathbf{a}_{ij} \tag{4}$$

The initialization of $\mathbf{h}^{(0)}$ was based on the assumption that the first coordinate of the embeddings corresponds to the identity relation. Accordingly, we want to ensure that $\phi((1, 0, \ldots, 0), (r_1, \ldots, r_n)) = (r_1, \ldots, r_n)$. We thus fix $\mathbf{a}_{1j} = \text{one-hot}(j)$, where we write one-hot$(j)$ to denote the $n$-dimensional vector which is 1 in the $j^{\text{th}}$ coordinate and 0 elsewhere. Note that the composition of two primitive relations is also a probability distribution over primitive relations. This provides an important inductive bias, as it encodes that the relationship between any two entities can be described by one of the $n$ considered primitive relations. Rule based methods, including NTP based approaches, also encode this assumption. However, the message passing operations that are used by standard GNNs typically do not. We hypothesise that the lack of this inductive bias partially explains why standard GNNs fail at systematic generalization for reasoning in relational domains.

**Pooling**  The pooling operator $\psi$ has to be chosen in accordance with the view of embeddings as epistemic states: if $\mathbf{x_1}, \ldots, \mathbf{x_k}$ capture sets of possible relationships then $\psi\{\mathbf{x_1}, \ldots, \mathbf{x_k}\}$ should intuitively capture the intersection of these sets. This requirement was studied by (Schockaert, 2024), whose central result is that the minimum and component-wise (i.e. Hadamard) product are compatible with this view, but sum pooling is not. In our setting, since the embeddings capture probability distributions rather than sets, $\psi$ is $L_1$-normalized after applying the minimum or product.

**Training**  To train the model, we assume that we have access to a set of examples of the form $(\mathcal{F}_i, h_i, t_i, r_i)$, where $h_i, t_i$ are entities appearing in $\mathcal{F}_i$ and the atom $r_i(h_i, t_i)$ can be inferred from $\mathcal{F}_i$. Different examples will involve a different fact set $\mathcal{F}_i$ but the set of relations $\mathcal{R}$ appearing in these fact sets is fixed. We train the model via contrastive learning. This means that for each

positive example $(\mathcal{F}_i, h_i, t_i, r_i)$ we can consider negative examples of the form $(\mathcal{F}_i, h_i, t_i, r')$ with $r' \in \mathcal{R} \setminus \{r_i\}$. Let us write $\mathbf{t}_i$ for the final-layer embedding of entity $t_i$ in the graph associated with $\mathcal{F}_i$ (with $h_i$ as the designated head entity). We train the model using a margin loss, imposing that the cross-entropy between $\mathbf{t}_i$ and $\mathbf{r}$ should be lower for positive examples than for negative examples.

**Facets** A crucial hyperparameter is the dimensionality $n$ of the embedding $\mathbf{r}$, as it determines the number of primitive relations. A large $n$ is essential to express all the relations of interest, but choosing $n$ too high may lead to overfitting. To address this, we propose to jointly train $m$ different models, each with a relatively low dimensionality. The underlying intuition is that each model focuses on a different *facet* of the relations. Because these facets are easier to model than the target relations themselves, each of the $m$ models can individually remain relatively simple. In the loss function, we simply add up the cross-entropies from these $m$ models (see Appendix F for details).

**Expressivity** The algebraic closure algorithm maintains a set of possible relations for each pair of entities. Simulating this algorithm thus requires a number of vectors which is quadratic in the number of entities. As this limits scalability, we instead consider an approximation, which we call *directional algebraic closure*, where we only maintain sets of possible relations between the head entity and the other entities (see Appendix C for details). EpiGNNs can be seen as a differentiable counterpart of directional algebraic closure, where standard directional algebraic closure emerges as a special case.

**Proposition 2** (informal). *There exists a parameterisation of the vectors $\mathbf{r}$ and $\mathbf{a}_{ij}$ such that the predictions of the EpiGNN exactly capture what can be inferred using directional algebraic closure.*

**Forward-backward model** As we noted in Section 2.1, the order in which the relations on a relational path $r_1; ...; r_k$ are composed sometimes matters. The model we have constructed thus far always composes such paths from left to right, i.e. we first compose $r_1$ and $r_2$, then compose the resulting relation with $r_3$, etc. To mitigate this limitation, we introduce a backward model, which relies on a designated tail entity. Similar as before, we initialise the embeddings as $\mathbf{t}^{(0)} = (1, 0 \ldots, 0)$ and $\mathbf{e}^{(0)} = (1/n, \ldots, 1/n)$ for $e \neq t$. These embeddings are updated as follows:

$$\mathbf{e}^{(l)} = \psi(\{\mathbf{e}^{(l-1)}\} \cup \{\phi(\mathbf{r}, \mathbf{f}^{(l-1)}) \,|\, r(e, f) \in \mathcal{F}\}) \tag{5}$$

where $\psi$ and $\phi$ are defined as before. Note that the backward model does not introduce any new parameters. We rely on the idea that $\phi$ captures the composition of relation vectors. In the forward model, the embedding of an entity $e$ is interpreted as capturing the relationship between $h$ and $e$ and in the backward model, it captures the relationship between $e$ and $t$. This is why $\mathbf{r}$ appears as the second argument of $\phi$ in (4) and as the first argument in (5). Let $\mathbf{e}^{\rightarrow}$ and $\mathbf{e}^{\leftarrow}$ be the final-layer embedding of $e$ in the forward and backward model. In particular, $\mathbf{t}^{\rightarrow}$ and $\mathbf{h}^{\leftarrow}$ now both capture the relationship between $h$ and $t$. Moreover, for any entity $e$ on a path between $h$ and $t$, the vector $\phi(\mathbf{e}^{\rightarrow}, \mathbf{e}^{\leftarrow})$ should also capture this relationship. We take advantage of the aggregation operator $\psi$ to take into account all these predictions. In particular, we construct the following vector:

$$\mathbf{s} = \psi(\{\mathbf{t}^{\rightarrow}, \mathbf{h}^{\leftarrow}\} \cup \{\phi(\mathbf{e}^{\rightarrow}, \mathbf{e}^{\leftarrow}) \,|\, e \in \mathcal{E}_{h,t}\}) \tag{6}$$

where we write $\mathcal{E}_{h,t}$ for the entities that appear on some path from $h$ to $t$. When there are multiple paths between $h$ and $t$, we randomly select one of the shortest paths to define $\mathcal{E}_{h,t}$. The model is then trained as before, using $\mathbf{s}$ as the predicted relation vector rather than $\mathbf{t}^{\rightarrow}$. Note that while this approach cannot exhaustively consider all possible derivations, it has the key advantage of remaining highly efficient. A schematic of the EpiGNN learning dynamics is shown in Figure 3.

## 4 EXPERIMENTS

We use the challenging problem of inductive relational reasoning to evaluate our proposed model against GNN, transformer and neuro-symbolic baselines. We consider two variants of the EpiGNN, which differ in the choice of the pooling operator: component-wise multiplication (EpiGNN-`mul`) and min-pooling (EpiGNN-`min`). Most of the considered benchmarks involve relation classification queries of the form $(h, ?, t)$, asking which relation holds between a given head entity $h$ and tail entity $t$. We focus in particular on systematic generalization, to assess whether models can deal with large distributional shifts from the training set, which is paramount in many real-world settings (Koh et al., 2021). We consider two existing benchmarks designed to test systematic generalization for relational

Table 1: Results (accuracy) on CLUTRR after training on problems with $k \in \{2, 3, 4\}$ and then evaluating on problems with $k \in \{5, \ldots, 10\}$. Results marked with $*$ were taken from (Minervini et al., 2020b), those with $\dagger$ from (Lu et al., 2022) and those with $2$ from (Cheng et al., 2023). The best performance for each $k$ is highlighted in **bold**.

|  | 5 Hops | 6 Hops | 7 Hops | 8 Hops | 9 Hops | 10 Hops |
|---|---|---|---|---|---|---|
| EpiGNN-`mul` (ours) | 0.99±.01 | **0.99±.01** | **0.99±.02** | 0.99±.03 | 0.96±.03 | **0.98±.02** |
| EpiGNN-`min` (ours) | 0.99±.01 | 0.98±.02 | 0.98±.03 | 0.97±.06 | 0.95±.04 | 0.93±.07 |
| NCRL$^2$ | **1.0±.01** | **0.99±.01** | 0.98±.02 | 0.98±.03 | 0.98±.03 | 0.97±.02 |
| R5$^\dagger$ | 0.99±.02 | 0.99±.04 | 0.99±.03 | **1.0±.02** | **0.99±.02** | 0.98±.03 |
| CTP$^*_L$ | 0.99±.02 | 0.98±.04 | 0.97±.04 | 0.98±.03 | 0.97±.04 | 0.95±.04 |
| CTP$^*_A$ | 0.99±.04 | 0.99±.03 | 0.97±.03 | 0.95±.06 | 0.93±.07 | 0.91±.05 |
| CTP$^*_M$ | 0.98±.04 | 0.97±.06 | 0.95±.06 | 0.94±.08 | 0.93±.08 | 0.90±.09 |
| GNTP$^*$ | 0.68±.28 | 0.63±.34 | 0.62±.31 | 0.59±.32 | 0.57±.34 | 0.52±.32 |
| ET | 0.99±.01 | 0.98±.02 | **0.99±.02** | 0.96±.04 | 0.92±.07 | 0.92±.07 |
| GAT$^*$ | 0.99±.00 | 0.85±.04 | 0.80±.03 | 0.71±.03 | 0.70±.03 | 0.68±.02 |
| GCN$^*$ | 0.94±.03 | 0.79±.02 | 0.61±.03 | 0.53±.04 | 0.53±.04 | 0.41±.04 |
| NBFNet | 0.83±.11 | 0.68±.09 | 0.58±.10 | 0.53±.07 | 0.50±.11 | 0.53±.08 |
| R-GCN | 0.97±.03 | 0.82±.11 | 0.60±.13 | 0.52±.11 | 0.50±.09 | 0.45±.09 |
| RNN$^*$ | 0.93±.06 | 0.87±.07 | 0.79±.11 | 0.73±.12 | 0.65±.16 | 0.64±.16 |
| LSTM$^*$ | 0.98±.03 | 0.95±.04 | 0.89±.10 | 0.84±.07 | 0.77±.11 | 0.78±.11 |
| GRU$^*$ | 0.95±.04 | 0.94±.03 | 0.87±.08 | 0.81±.13 | 0.74±.15 | 0.75±.15 |

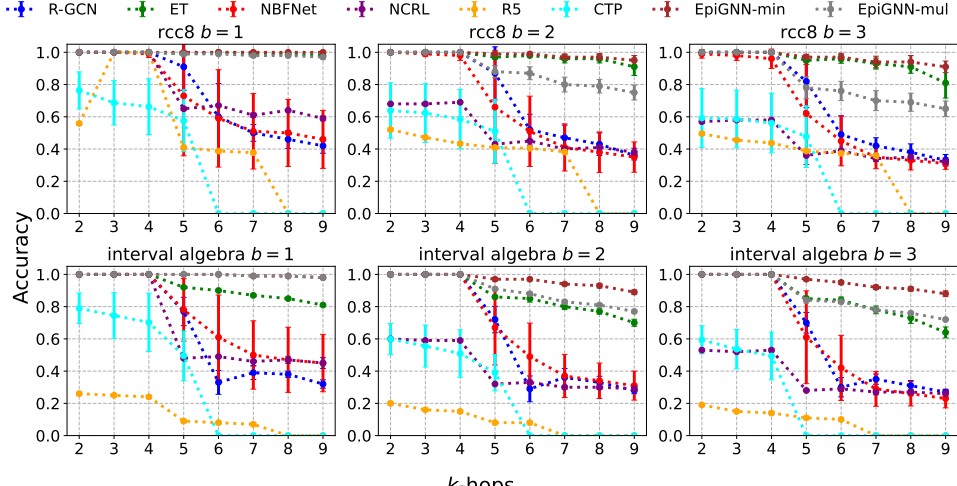

Figure 4: RCC-8 and Interval Algebra benchmark results (accuracy). R5 and CTP results for 5+ hops were set to zero since the model took longer than 30 minutes for inference. Models are trained on graphs with $b \in \{1, 2, 3\}$ paths of length $k \in \{2, 3, 4\}$. The best model for all cases is EpiGNN-`min`.

reasoning: CLUTRR (Sinha et al., 2019) and Graphlog (Sinha et al., 2020). We also evaluate on two novel benchmarks: one involving RCC-8 relations and one based on IA. These go beyond existing benchmarks on two fronts, as illustrated in Figure 1: (i) the need to go beyond Horn rules to capture relational compositions and (ii) requiring models to aggregate information multiple relational paths. For CLUTRR, RCC-8 and IA, to test for systematic generalization, models are trained on small graphs and subsequently evaluated on larger graphs. In particular, for CLUTRR, the length $k$ of the considered relational paths is varied, while for RCC-8 and IA we vary both the number of relational paths $b$ and their length $k$. In the case of Graphlog, the size of training and test graphs is similar, but models still need to apply learned rules in novel ways to perform well. We complement these experiments with an evaluation on the popular task of inductive knowledge graph completion. Here, the need for systematic generalization is less obvious and a much broader family of methods can be used. The main purpose of this analysis is to analyze how well EpiGNNs perform compared to domain-specialized models in a more general setting. Finally, we also analyze the parameter and time complexity of EpiGNNs.

Table 2: Results on Graphlog (accuracy). For each world, we report the number of distinct relation sequences between head and tail (ND) and the Average resolution length (ARL). Results marked with * were taken from (Lu et al., 2022) and those with † from (Cheng et al., 2023). The best and second-best performance across all the models are highlighted in **bold** or underlined.

| World ID | ND | ARL | E-GAT* | R-GCN* | CTP* | R5* | NCRL† | ET | EpiGNN-`mul` |
|---|---|---|---|---|---|---|---|---|---|
| World 6 | 249 | 5.06 | 0.536 | 0.498 | 0.533±0.03 | 0.687±0.05 | **0.702**±0.02 | 0.496 ± 0.087 | 0.648 ± 0.012 |
| World 7 | 288 | 4.47 | 0.613 | 0.537 | 0.513±0.03 | **0.749**±0.04 | - | 0.487 ± 0.056 | 0.611±0.026 |
| World 8 | 404 | 5.43 | 0.643 | 0.569 | 0.545±0.02 | 0.671±0.03 | **0.687**±0.02 | 0.55 ± 0.092 | 0.649±0.042 |
| World 11 | 194 | 4.29 | 0.552 | 0.456 | 0.553±0.01 | **0.803**±0.01 | - | 0.637 ± 0.091 | 0.758 ± 0.037 |
| World 32 | 287 | 4.66 | 0.700 | 0.621 | 0.581±0.04 | 0.841±0.03 | - | 0.815 ± 0.061 | **0.914**±0.026 |

Table 3: Hits@10 results on the inductive benchmark datasets extracted from WN18RR, FB15k-237 with 50 negative samples. The results of other baselines except NBFNet are obtained from (Liu et al., 2023) and the former from (Zhu et al., 2021). The best and second-best performance across all models are highlighted in **bold** or underlined.

| | | WN18RR | | | | FB15k-237 | | | |
|---|---|---|---|---|---|---|---|---|---|
| | | v1 | v2 | v3 | v4 | v1 | v2 | v3 | v4 |
| Rule-Based | Neural LP | 74.37 | 68.93 | 46.18 | 67.13 | 52.92 | 58.94 | 52.90 | 55.88 |
| | DRUM | 74.37 | 68.93 | 46.18 | 67.13 | 52.92 | 58.73 | 52.90 | 55.88 |
| | RuleN | 80.85 | 78.23 | 53.39 | 71.59 | 49.76 | 77.82 | 87.69 | 85.60 |
| Graph-Based | GraIL | 82.45 | 78.68 | 58.43 | 73.41 | 64.15 | 81.80 | 82.83 | 89.29 |
| | CoMPILE | 83.60 | 79.82 | 60.69 | 75.49 | 67.64 | 82.98 | 84.67 | 87.44 |
| | TACT | 84.04 | 81.63 | 67.97 | 76.56 | 65.76 | 83.56 | 85.20 | 88.69 |
| | SNRI | 87.23 | 83.10 | 67.31 | 83.32 | 71.79 | 86.50 | 89.59 | 89.39 |
| | ConGLR | 85.64 | 92.93 | 70.74 | 92.90 | 68.29 | 85.98 | 88.61 | 89.31 |
| | REST | **96.28** | **94.56** | 79.50 | **94.19** | 75.12 | 91.21 | 93.06 | **96.06** |
| | NBFNet | 94.80 | 90.50 | **89.30** | 89.00 | 83.40 | 94.90 | **95.10** | 96.00 |
| | EpiGNN-`min` | 92.45 | 85.99 | 84.18 | 85.77 | **91.67** | **95.54** | 93.74 | 93.45 |

**Main results** Results for CLUTRR, RCC-8, IA and Graphlog are shown in Table 1, Figure 4 and Table 2. We report the average accuracy and $2\sigma$ errors across 10 seeds for CLUTRR and 3 seeds for RCC-8, IA and Graphlog. For CLUTRR, both variants of our model outperform all GNN and RNN methods, as well as edge transformers (ET). The EpiGNN-`mul` model is also on par with the SOTA neuro-symbolic methods NCRL (Cheng et al., 2023) and R5 (Lu et al., 2022). For RCC-8 and IA, as expected, the neuro-symbolic methods are largely ineffective, being substantially outperformed by our method as well as by ET. This highlights the fact that neuro-symbolic methods are not capable of modeling disjunctive rules. Our model with min-pooling achieves the best results. Edge transformers also perform well, especially for RCC-8, but they underperform on the most challenging configurations (e.g. $k = 9$ and $b = 3$). Comparing the `mul` and `min` variants of our model, we find that `mul` performs better on single-path problems (CLUTRR and Graphlog), while `min` leads to better results when paths need to be aggregated (RCC-8 and IA). For Graphlog, we only consider worlds that are characterized as 'hard' by Lu et al. (2022), with an Average Resolution Length (ARL) $> 4$. Graphlog is amenable to path-based reasoning but is noisier than the other datasets. We find that EpiGNN-`mul` is outperformed by R5 and NCRL in most cases, thanks to their stronger inductive bias. However, EpiGNN-`mul` clearly improves the SOTA for World 32. Moreover, EpiGNN-`mul` outperforms CTPs and the GNN baselines.

The results for inductive knowledge graph completion (KGC) are summarized in Table 3. This task involves link prediction queries of the form $(h, r, ?)$, asking for tail entities that are in relation $r$ with some head entity $h$. We use the inductive splits of FB15k-237 (Toutanova & Chen, 2015) and WN18RR (Dettmers et al., 2018b) from by Teru et al. (2020) and the evaluation protocol of Bordes et al. (2013). In inductive KGC, models are evaluated on a test graph which is disjoint from the training graph. Models thus need to learn inference patterns, but they may not need to systematically generalize them. Note in particular that the inference chains that are needed for the training and test graphs come from the same distribution and have the same length. Since all entities need to be scored, we use the forward-only EpiGNN to efficiently score all entities in the knowledge graph with respect to the query relation $r$. The results show that EpiGNNs perform well, being competitive with SOTA models, and even achieving the best results in two cases, despite not being designed for this task.

**Ablations** We confirm the importance of key architectural components of our model through an ablation study on CLUTRR and RCC-8. To show the importance of jointly training multiple lower-dimensional models, we show results for for a variant with only $m = 1$ facet. To show the importance of modeling embeddings as epistemic states, we test a variant in which we remove the requirement that embedding components are non-negative and sum to 1. To show the importance of the bilinear composition function $\phi$ in (4), we test a variant where the composition function is replaced by distmul (Yang et al., 2015) after applying a 4-layer MLP with ReLU activation to both inputs of $\phi$ (with the MLP being added to ensure the model is not underparameterised). To show the importance of the foward-backward nature of the model, we test a variant in which only the forward embeddings $\mathbf{t}^{\rightarrow}$ are used. The results in Table 4 confirm the importance of all these components, with the use of embeddings as epistemic states and the bilinear form of $\phi$ being particularly important.

Table 4: Ablation study results. We show the average accuracy across all configurations (i.e. 4-10 hops for CLUTRR, and $k \in \{2, ..., 9\}$ and $b \in \{1, 2, 3\}$ for RCC-8), and the accuracy for the hardest setting (i.e. 10 hops for CLUTRR, and $b = 3$, $k = 9$ for RCC-8).

|  | CLUTRR | | RCC-8 | |
|---|---|---|---|---|
|  | **Avg** | **Hard** | **Avg** | **Hard** |
| EpiGNN | 0.99 | 0.99 | 0.96 | 0.80 |
| - With facets=1 | 0.94 | 0.85 | 0.92 | 0.68 |
| - Unconstrained embeddings | 0.36 | 0.30 | 0.38 | 0.21 |
| - MLP+distmul composition | 0.29 | 0.31 | 0.13 | 0.13 |
| - Forward model only | 0.94 | 0.82 | 0.84 | 0.51 |

**Parameter and time complexity** Writing $n$ for the total dimensionality of the relation vectors and $m$ for the number of facets, the number of parameters is exactly $|\mathcal{R}|n + \frac{n^3}{m^2}$, namely $|\mathcal{R}|n$ parameters for encoding the relation vectors and $\frac{n^3}{m^3}$ parameters for encoding the $\mathbf{a}_{ij}$ vectors in each of the $m$ facets. In practice, we use a large number of facets, which allows us to keep $\frac{n}{m}$ small. In particular, when comparing our model with existing neuro-symbolic models, after hyperparameter tuning each of the models, we find that EpiGNNs are at least two orders of magnitude smaller, as shown in Figure 5. The time complexity of the EpiGNN is $O(|\mathcal{E}|n + |\mathcal{F}|\frac{n^3}{m^2})$, and thus highly efficient given that $\frac{n}{m}$ is small in practice. A comparison with the main baselines is shown in Appendix G.5.

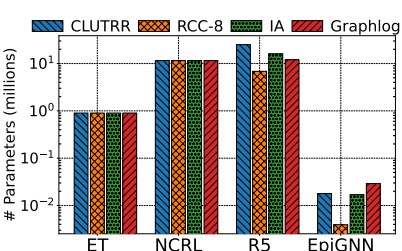

Figure 5: Parameter complexity across the relation prediction benchmarks.

# 5 RELATED WORK

**Neuro-symbolic methods** Neuro-symbolic methods such as DeepProbLog (Manhaeve et al., 2018) and NTPs (Rocktäschel & Riedel, 2017) are essentially differentiable formulations of inductive logic programming, where the latter is concerned with learning symbolic rule bases from sets of examples (Muggleton, 1991; Muggleton & De Raedt, 1994; Nienhuys-Cheng & de Wolf, 1997; Evans & Grefenstette, 2018). Although these methods are capable of systematic reasoning, scalability is a key concern. For instance, by generalizing logic programming, NTPs and DeepProbLog rely on the exploration of a potentially exponential number of proof paths. Another challenge to scalability comes from the fact that the space of candidate rules grows exponentially with the number of relations (Evans & Grefenstette, 2018). More recent methods, including R5 (Lu et al., 2022), CTPs (Minervini et al., 2020b) and NCRL (Cheng et al., 2023) are somewhat more efficient than NTPs. R5 uses Monte Carlo Tree search with a dynamic rule memory network, CTPs use a learned heuristic filter function on top of NTPs to reduce the number of explored paths during backward chaining, and NCRL focuses on learning how to iteratively collapse relational paths. These methods can systematically reason but only NCRL scales to knowledge graphs such as FB15k. Moreover, both R5 and NCRL are limited by their over-reliance on single path sampling and cannot deal with disjunctive rules. CTPs cannot handle benchmarks such as RCC-8 and IA either, as they cannot model the constraint that the different relations are pairwise disjoint and jointly exhaustive. Some approaches, such as Logic Tensor Networks (Badreddine et al., 2022) and Lifted Relational Neural Networks (Sourek et al., 2018) rely on fuzzy logic connectives to enable more efficient differentiable logic programming. These frameworks are typically quite general, e.g. being capable of encoding GNNs as a special case (Sourek et al., 2021). As such, they should be viewed as modeling frameworks rather than specific models. We are not aware of work that uses these frameworks for systematic generalization.

More generally, within the context of statistical relational learning, methods such as Markov Logic Networks (Richardson & Domingos, 2006) have been studied, which can learn weighted sets of arbitrary propositional formulas, but suffer from limited scalability, a limitation which is inherited by differentiable versions of such methods (Marra & Kuželka, 2021).

**Systematic reasoning** There has been a large body of work on learning to reason with neural networks, including recent approaches based on GNNs (Zhang et al., 2020; Zhu et al., 2021; Zhang & Yao, 2022), pre-trained language models (Clark et al., 2020; Kojima et al., 2022; Creswell et al., 2023) or tailor-made architectures for relational reasoning (Santoro et al., 2017; Bergen et al., 2021). Most of these methods, however, fail at tasks that require systematic generalization (Sinha et al., 2019; 2020) and are prone learning reasoning shortcuts (Marconato et al., 2023; Zhang et al., 2023a). NBFNet (Zhu et al., 2021), R-GCN (Schlichtkrull et al., 2018), and graph-convolution methods (Dettmers et al., 2018a; Vashishth et al., 2020) are examples of scalable and (mostly) parameter-efficient methods designed for knowledge graph completion. Such methods typically leverage knowledge graph embedding methods (Bordes et al., 2013; Trouillon et al., 2017; Yang et al., 2015) in the message passing step. NBFNet uses such methods to compose relations and anchors the embeddings to a head entity to develop path-based relational representations. GNNs for logical reasoning have also been proposed, including R2N (Marra et al., 2023) and LERP (Han et al., 2023), but their focus is again on knowledge graphs and not systematic reasoning. For instance, R2N uses multi-layer perceptrons (MLPs) for information composition and sum aggregation. This increases the capacity of the model, which can be beneficial for learning statistical regularities from knowledge graphs, but which at the same time also hurts a GNN's systematic reasoning ability, as we have seen in our ablation analysis. EpiGNNs are inspired by the idea of algorithmic alignment (Xu et al., 2020), i.e. aligning the neural architecture with the desired reasoning algorithm. Edge Transformers (Bergen et al., 2021) also rely on this idea to some extent, aligning the model with relational reasoning by using a triangular variant of attention (Vaswani et al., 2017) to capture relational compositions. However, compared to EpiGNNs, edge transformers are less scalable, as they cannot operate on sparse graphs, less parameter-efficient, and less successful at systematic generalization, as we have seen in our experiments. Finally, the problem of systematic generalization has also been studied beyond relational reasoning. For instance, there is existing work on benchmarking length generalization for sequence based methods such as pretrained transformers and recurrent networks, including SCAN (Lake & Baroni, 2018), LEGO (Zhang et al., 2023b) and Addition (Nye et al., 2021).

## 6 CONCLUSIONS

We have challenged the view that Graph Neural Networks are not capable of systematic generalization in relational reasoning problems, by introducing the EpiGNN, a principled GNN architecture for this setting. To impose an appropriate inductive bias, node embeddings in our framework are treated as epistemic states, intuitively capturing sets of possible relationships, which are iteratively refined by the EpiGNN. In this way, EpiGNNs are closely aligned with the algebraic closure algorithm, which is used for relational reasoning by symbolic methods, and a formal connection with this algorithm has been established. EpiGNNs are scalable and parameter-efficient. They rival neuro-symbolic methods on standard systematic reasoning benchmarks such as CLUTRR and Graphlog, while clearly outperforming existing GNN and transformer based methods. Moreover, we have highlighted that existing neuro-symbolic methods have an important weakness, which arises from an implicit assumption that relations can be predicted from a single relational path. To explore this issue, we have introduced two new benchmarks based on spatio-temporal calculi, finding that neuro-symbolic methods indeed fail on them, while EpiGNNs still performs well in this more challenging setting. Finally, despite being designed for systematic reasoning, our experiments show that EpiGNNs rival SOTA specialized methods for inductive knowledge graph completion.

**Limitations** The EpiGNN is limited by its statistical nature, where the parameters of the model will inevitably be biased by training data artefacts. While we have shown that our model can perform well in settings where existing neuro-symbolic methods fail (i.e. spatio-temporal benchmarks), the much stronger inductive bias imposed by the latter puts them at an advantage in severely data-limited settings. This can be seen in some of the Graphlog results. For a variant of CLUTRR where the model is only trained on problems of size $k \in \{2, 3\}$, our model is also outperformed by the best neuro-symbolic methods (shown in Appendix G).

ACKNOWLEDGEMENTS

This work was supported by the EPSRC grant EP/W003309/1.

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

## A   SEMANTICS OF ENTAILMENT

For simple path rules of the form (1), the semantics of entailment can be defined in terms of the immediate consequence operator. Let $\mathcal{K}_{\mathcal{F}}$ be the grounding of $\mathcal{K}$ w.r.t. $\mathcal{F}$, i.e. for each rule of the form $r(X,Z) \leftarrow r_1(X,Y) \wedge r_2(Y,Z)$ in $\mathcal{K}$, $\mathcal{K}_{\mathcal{F}}$ contains all the possible rules of the form $r(a,c) \leftarrow r_1(a,b) \wedge r_2(b,c)$ that can be obtained by substituting the variables for entities that appear in $\mathcal{F}$. Since we assume that both $\mathcal{F}$ and $\mathcal{K}$ are finite, we have that $\mathcal{K}_{\mathcal{F}}$ is finite as well. Let $\mathcal{F}_0 = \mathcal{F}$ and for $i \geq 1$ define:

$$\mathcal{F}_i = \mathcal{F}_{i-1} \cup \{r(a,c) \mid \exists b \in \mathcal{E} \,.\, r_1(a,b), r_2(b,c) \in \mathcal{F}_{i-1} \text{ and } (r(a,c) \leftarrow r_1(a,b) \wedge r_2(b,c)) \in \mathcal{K}_{\mathcal{F}}\}$$

Since there are finitely many atoms $r(a,c)$ that can be constructed using the entities and relations appearing in $\mathcal{K}_{\mathcal{F}}$, this process reaches a fixpoint after a finite number of steps, i.e. for some $\ell \in \mathbb{N}$ we have $\mathcal{F}_\ell = \mathcal{F}_{\ell+1}$. Let us write $\mathcal{F}^*$ for this fixpoint. Then we define $\mathcal{F} \cup \mathcal{K} \models r(a,b)$ iff $r(a,b) \in \mathcal{F}^*$.

**Example 1.** *The canonical example for this setting concerns reasoning about family relationships. In this case, $\mathcal{K}$ contains rules such as:*

$$grandfather(X,Z) \leftarrow father(X,Y) \wedge mother(Y,Z)$$

*If $\mathcal{F} = \{father(bob, alice), mother(alice, eve)\}$ then $\mathcal{K}_{\mathcal{F}}$ contains rules such as:*

$$grandfather(bob, eve) \leftarrow father(bob, alice) \wedge mother(alice, eve)$$
$$grandfather(bob, alice) \leftarrow father(bob, alice) \wedge mother(alice, alice)$$
$$grandfather(bob, bob) \leftarrow father(bob, alice) \wedge mother(alice, bob)$$
$$grandfather(bob, eve) \leftarrow father(bob, eve) \wedge mother(eve, eve)$$
$$\dots$$

*We have $\mathcal{F}_1 = \{father(bob, alice), mother(alice, eve), grandfather(bob, eve)\}$, with $\mathcal{F}_1 = \mathcal{F}_2 = \mathcal{F}^*$. We thus find: $\mathcal{K} \cup \mathcal{F} \models grandfather(bob, eve)$.*

For the more general setting with disjunctive rules and constraints, we can define the semantics of entailment in terms of Herbrand models. Given a set of disjunctive rules and constraints $\mathcal{K}$ and a set of facts $\mathcal{F}$, the Herbrand universe $\mathcal{U}_{\mathcal{K},\mathcal{F}}$ is the set of all atoms of the form $r(a,b)$ which can be constructed from a relation $r$ and entities $a, b$ appearing in $\mathcal{K} \cup \mathcal{F}$. A Herbrand interpretation $\omega$ is a subset of $\mathcal{U}_{\mathcal{K},\mathcal{F}}$. The interpretation $\omega$ satisfies a ground disjunctive rule of the form $s_1(a,c) \vee \dots \vee s_k(a,c) \leftarrow r_1(a,b) \wedge r_2(b,c)$ iff either $\{r_1(a,b), r_2(b,c)\} \not\subseteq \omega$ or $\omega \cap \{s_1(a,c), \dots, s_k(a,c)\} \neq \emptyset$. The interpretation $\omega$ satisfies a ground constraint of the form $\bot \leftarrow r_1(a,b) \wedge r_2(a,b)$ iff $\{r_1(a,b), r_2(a,b)\} \not\subseteq \omega$. We say that $\omega$ is a model of the grounding $\mathcal{K}_{\mathcal{F}}$ iff $\omega$ satisfies all the ground rules and constraints in $\mathcal{K}_{\mathcal{F}}$. Finally, we have that $\mathcal{K} \cup \mathcal{F} \models r(a,b)$ iff $r(a,b)$ is contained in every model of $\mathcal{K}_{\mathcal{F}}$.

## B   SIMPLE PATH ENTAILMENT: PROOF OF PROPOSITION 1

The correctness of Proposition 1 immediately follows from Lemmas 1 and 2 below.

**Lemma 1.** *Suppose there exists a relational path $r_1; \dots; r_k$ connecting $a$ and $b$ in $G_{\mathcal{F}}$ such that $r$ can be derived from $r_1; \dots; r_k$. Then it holds that $\mathcal{K} \cup \mathcal{F} \models r(a,b)$.*

*Proof.* Let $r_1; \dots; r_k$ be a relational path connecting $a$ and $b$, such that $r$ can be derived from $r_1; \dots; r_k$. Then $\mathcal{F}$ contains facts of the form $r_1(a, x_1), r_2(x_1, x_2), \dots, r_k(x_{k-1}, b)$. Let us now consider the derivation from $r_1; \dots; r_k$ to $r$. After the first derivation step, we have a relational path of the form $r_1; \dots; r_{i-2}; s; r_{i+1}; \dots; r_k$. In this case, $\mathcal{K}$ contains a rule of the form $s(X,Z) \leftarrow r_{i-1}(X,Y) \wedge r_i(Y,Z)$. This means that $\mathcal{K} \cup \mathcal{F} \models s(x_{i-2}, x_i)$. We thus have a relational path of the form $r_1; \dots; r_{i-2}; s; r_{i+1}; \dots; r_k$, where each relation is associated with an atom that is entailed by $\mathcal{K} \cup \mathcal{F}$. Each derivation step introduces such an atom (while reducing the length of the relational path by 1). After $k - 1$ steps, we thus obtain the atom $r(a,b)$, from which it follows that $\mathcal{K} \cup \mathcal{F} \models r(a,b)$.   $\square$

**Lemma 2.** *Suppose that $\mathcal{K} \cup \mathcal{F} \models r(a,b)$. Then it holds that there exists a relational path $r_1; \dots; r_k$ connecting $a$ and $b$ in $G_{\mathcal{F}}$ such that $r$ can be derived from $r_1; \dots; r_k$.*

*Proof.* Recall from Appendix A that $\mathcal{K} \cup \mathcal{F} \models r(a, b)$ iff $r(a, b) \in \mathcal{F}^*$. It thus suffices to show by induction that $r(a, b) \in \mathcal{F}_i$ implies that there exists a relational path $r_1; \ldots; r_k$ connecting $a$ and $b$ in $G_{\mathcal{F}}$ such that $r$ can be derived from $r_1; \ldots; r_k$. First, if $r(a, b) \in \mathcal{F}_0$ then by definition there is a relational path $r$ connecting $a$ and $b$ in $G_{\mathcal{F}}$, meaning that the result is trivially satisfied. Now suppose that the result has already been shown for $\mathcal{F}_{i-1}$. Let $r(a, b) \in \mathcal{F}_i \setminus \mathcal{F}_{i-1}$. Then there must exist facts $r_1(a, x)$ and $r_2(x, b)$ in $\mathcal{F}_{i-1}$ such that $\mathcal{K}$ contains a rule of the form $r(X, Z) \leftarrow r_1(X, Y) \wedge r_2(Y, Z)$. By induction, we have that there is a relational path $s_1; \ldots; s_{\ell_1}$ connecting $a$ and $x$ such that $r_1$ can be derived from this path, and a relational path $t_1; \ldots; t_{\ell_2}$ connecting $x$ and $b$ from which $r_2$ can be derived. We then have that $s_1; \ldots s_{\ell_1}; t_1; \ldots; t_{\ell_2}$ is a path connecting $a$ and $b$. Furthermore, we have that $r_1; r_2$ can be derived from this path, and thus also $r$. $\square$

## C  REASONING ABOUT DISJUNCTIVE RULES USING ALGEBRAIC CLOSURE

We consider knowledge bases with three types of rules. First, we have disjunctive rules that encode relational compositions:

$$s_1(X, Z) \vee \ldots \vee s_k(X, Z) \leftarrow r_1(X, Y) \wedge r_2(Y, Z) \tag{7}$$

Second, we have constraints of the following form:

$$\bot \leftarrow r_1(X, Y) \wedge r_2(X, Y) \tag{8}$$

meaning that $r_1$ and $r_2$ are disjoint. Finally, we consider knowledge about inverse relations, expressed using rules of the following form:

$$r_2(Y, X) \leftarrow r_1(X, Y) \tag{9}$$

We now describe the algebraic closure algorithm, which can be used to decide entailment for calculi such as RCC-8 and IA. We also describe an approximation of this algorithm, which we call *directional algebraic closure*.

**Full algebraic closure**   Let us make the following assumptions:

- The knowledge base $\mathcal{K}$ contains rules of the form (7), encoding the composition of relations $r_1$ and $r_2$.
- We also have that $\mathcal{K}$ contains the rule $\bigvee_{r \in \mathcal{R}} r(X, Y) \leftarrow \top$, expressing that the set of relations is exhaustive.
- For all distinct relations $r_1, r_2 \in \mathcal{R}$, $r_1 \neq r_2$, $\mathcal{K}$ contains a constraint of the form (8), expressing that the relations are pairwise disjoint.
- For every $r \in \mathcal{R}$ there is some relation $\hat{r} \in \mathcal{R}$, such that $\mathcal{K}$ contains the rules $r(Y, X) \leftarrow \hat{r}(X, Y)$ and $\hat{r}(Y, X) \leftarrow r(X, Y)$, expressing that $\hat{r}$ is the inverse of $r$.
- $\mathcal{K}$ contains no other rules.

Let us write $r_1 \circ r_2$ for the set of relations that appears in the head of the rule defining the composition of $r_1$ and $r_2$ in $\mathcal{K}$. If no such rule exists in $\mathcal{K}$ for $r_1$ and $r_2$ then we define $r_1 \circ r_2 = \mathcal{R}$. Let $\mathcal{E}$ be the set of entities that appear in $\mathcal{F}$. Let us assume that $\mathcal{F}$ is consistent with $\mathcal{K}$, i.e. $\mathcal{K} \cup \mathcal{F} \not\models \bot$.

The main idea of the algebraic closure algorithm is that we iteratively refine our knowledge of what relationships are possible between different entities. Specifically, for all entities $e, f \in \mathcal{E}$, we define the initial set of possible relationships as follows:

$$X_{ef}^{(0)} = \begin{cases} \{r\} & \text{if } r(e, f) \in \mathcal{F} \\ \{\hat{r}\} & \text{if } r(f, e) \in \mathcal{F} \\ \mathcal{R} & \text{otherwise} \end{cases}$$

where $\hat{r}$ is the unique relation that is asserted to be the inverse of $r$ in $\mathcal{K}$. Note that because we assumed $\mathcal{F}$ to be consistent, if $r(e, f) \in \mathcal{F}$ and $r'(f, e) \in \mathcal{F}$ it must be the case that $r' = \hat{r}$. We now iteratively refine the sets $X_{ef}^{(i)}$. Specifically, for $i \geq 1$, we define the refinement step:

$$X_{ef}^{(i)} = X_{ef}^{(i-1)} \cap \bigcap \{X_{eg}^{(i-1)} \diamond X_{gf}^{(i-1)} \mid g \in \mathcal{E}\} \tag{10}$$

where, for $X, Y \subseteq \mathcal{R}$, we define:

$$X \diamond Y = \bigcup \{r \circ s \mid r \in X, s \in Y\}$$

Since each of the sets $X_{ef}^{(i)}$ only contains finitely many elements, and there are only finitely many such sets, this process must clearly converge after a finite number of steps. Let us write $X_{ef}$ of the sets of relations that are obtained upon convergence. For many spatial and temporal calculi, we have that $r \in X_{ef}$ iff $\mathcal{K} \cup \mathcal{F} \cup \{r(e, f)\} \not\models \bot$. In other words, $X_{ef}$ encodes everything that can be inferred about the relationship between $e$ and $f$. In particular, we have have $\mathcal{K} \cup \mathcal{F} \models r(e, f)$ iff $X_{ef} = \{r\}$. Note that this equivalence only holds for particular calculi: in general, $\mathcal{K} \cup \mathcal{F} \models r(e, f)$ does not imply $X_{ef} = \{r\}$. Furthermore, even for RCC-8 and IA, this equivalence only holds because the initial set of facts $\mathcal{F}$ is disjunction-free. We refer to (Renz, 1999; Krokhin et al., 2003; Broxvall et al., 2002) for more details on when algebraic closure decides entailment.

**Directional algebraic closure**   The algebraic closure algorithm relies on sets $X_{ef}^{(i)}$ for every pair of entities, which limits its scalability (although more efficient special cases have been studied (Amaneddine et al., 2013)). Any simulation of this algorithm using a neural network would thus be unlikely to scale to large graphs. For this reason, we study an approximation, which we refer to as *directional algebraic closure*. This approximation essentially aims to infer the possible relationships between a fixed head entity $h$ and all the other entities from the graph. The relationship between $h$ and a given entity $e$ is determined based on the paths in $\mathcal{G}$ connecting $h$ to $e$. For this approximation, we furthermore omit rules about inverse relations. Other than this, we make the same assumptions about $\mathcal{K}$ as before. We now learn sets $X_e^{(0)}$ which capture the possible relationships between $h$ and $e$. These sets are initialized as follows:

$$X_e^{(0)} = \begin{cases} \{r\} & \text{if } r(h, e) \in \mathcal{F} \\ \mathcal{R} & \text{otherwise} \end{cases}$$

We define for $i \geq 1$, the refinement step:

$$X_e^{(i)} = X_e^{(i-1)} \cap \bigcap \{X_f^{(i-1)} \diamond s \mid s(f, e) \in \mathcal{F}\} \tag{11}$$

with

$$X_f^{(i-1)} \diamond s = \bigcup \{r \circ s \mid r \in X_f^{(i-1)}\}$$

where $r_1 \circ r_2$ is defined as before. We write $X_e$ to denote the sets $X_e^{(i)}$ that are obtained upon convergence. Clearly, after a finite number of iterations $i$ we have $X_e^{(i)} = X_e^{(i+1)}$. Note how the directional closure algorithm essentially limits the full algebraic closure algorithm to compositions of the form $X_{he} \diamond X_{ef}$, for entities $e$ and $f$ such that there is a fact of the form $r(e, f)$ in $\mathcal{F}$. As the following result shows, the algorithm is still sound, although it may infer less knowledge than the full algebraic closure algorithm.

For a rule $\rho$ of the form (7), we write $head(\rho)$ to denote the set $\{s_1, ..., s_k\}$ of relations appearing in the head of the rule and $body(\rho)$ to denote the pair $(r_1, r_2)$ of relations appearing in the body.

**Proposition 3.** *Assume that $\mathcal{K}$ consists of (i) rules of the form (7), (ii) for each pair of distinct relations $r_1, r_2$ the disjointness constraint (8), and (iii) the rule $\bigvee_{r \in \mathcal{R}} r(X, Y) \leftarrow \top$. Let $e \in \mathcal{E}$. It holds that $\mathcal{K} \cup \mathcal{F} \models \bigvee_{r \in X_e} r(h, e)$.*

*Proof.* We show this result by induction. First note that we have $\mathcal{K} \cup \mathcal{F} \models \bigvee_{r \in X_e^{(0)}} r(h, e)$. Indeed, either $X_e^{(0)} = \{r(h, e)\}$ with $r(h, e) \in \mathcal{F}$, in which case the claim clearly holds, or $X_e^{(0)} = \mathcal{R}$, in which case the claim holds because $\mathcal{K}$ contains the rule $\bigvee_{r \in \mathcal{R}} r(X, Y) \leftarrow \top$. Now suppose we have already established $\mathcal{K} \cup \mathcal{F} \models \bigvee_{r \in X_e^{(i-1)}} r(h, e)$ for every $e \in \mathcal{E}$. To show that $\mathcal{K} \cup \mathcal{F} \models \bigvee_{r \in X_e^{(i)}} r(h, e)$, it is sufficient to show that $\mathcal{K} \cup \mathcal{F} \models \neg r(h, e)$ for every $r \in X_e^{(i-1)} \setminus X_e^{(i)}$. Let $r \in X_e^{(i-1)} \setminus X_e^{(i)}$. Then there must exist some $s(f, e) \in \mathcal{F}$ such that $r \notin X_f^{(i-1)} \diamond s$. In that case, for every $t \in X_f^{(i-1)}$ we have $r \notin t \circ s$. This means that for every $t \in X_f^{(i-1)}$ there exists some rule $\rho_t$ in $\mathcal{K}$ such that $body(\rho_t) = (t, s)$ and $r \notin heads(\rho_t)$. We have for every $t \in X_f^{(i-1)}$ that

$\mathcal{K} \cup \mathcal{F} \models t(h, f) \rightarrow \vee_{u \in heads(\rho_t)} u(h, e)$. Moreover, because of our assumption that $\mathcal{K}$ encodes that all relations are pairwise disjoint, for $u \neq r$ we have $\mathcal{K} \models u(h, e) \rightarrow \neg r(h, e)$. We thus find for every $t \in X_f^{(i-1)}$ that $\mathcal{K} \cup \mathcal{F} \models t(h, f) \rightarrow \neg r(h, e)$. By induction we also have $\mathcal{K} \cup \mathcal{F} \models \bigvee_{t \in X_f^{(i-1)}} t(h, e)$. Together we find $\mathcal{K} \cup \mathcal{F} \models \neg r(h, e)$. $\qquad \square$

## D  EXPRESSIVITY: PROOF OF PROPOSITION 2

In this section, we provide a proof for Proposition 2, which we first restate formally in Proposition 4. We will show that the base EpiGNN model (i.e. the forward model) is already capable of simulating the directional algebraic closure algorithm. The proof associates the embeddings $\mathbf{e}^{(i)}$ from the GNN with the sets $X_e^{(i)}$. We show the result for the min-pooling operator $\psi_{\min}$ defined as follows:

$$\psi_{\min}(\mathbf{x_1}, \ldots, \mathbf{x_k}) = \frac{\min(\mathbf{x_1}, \ldots, \mathbf{x_k})}{\|\min(\mathbf{x_1}, \ldots, \mathbf{x_k})\|_1} \tag{12}$$

**Proposition 4.** *Let $\mathcal{F}$ be a set of facts and $\mathcal{K}$ a knowledge base satisfying the conditions of Proposition 3. Furthermore assume that $\mathcal{K} \cup \mathcal{F}$ is consistent, i.e. $\mathcal{K} \cup \mathcal{F} \not\models \bot$. Let $X_e^{(i)}$ be sets of relations that are constructed using the directional algebraic closure algorithm, and let $e_j^i$ denote the $j^{th}$ coordinate of $\mathbf{e}^{(i)}$. Let the pooling operation be chosen as $\psi = \psi_{\min}$. There exists a parameterisation of the vectors $\mathbf{r}_i$ and $\mathbf{a}_{ij}$ such that the following refinement condition for (11) holds:*

$$X_e^{(i)} \supseteq \{r_j \mid e_j^{i+1} > 0, 2 \leq j \leq n\} \supseteq X_e^{(i+1)} \tag{13}$$

*Proof.* Let $r_1, ..., r_n$ be an enumeration of the relations in $\mathcal{R} \cup \{id\}$, where we fix $r_1 = id$. Let $\mathbf{a}_{ij} = (a_1^{ij}, ..., a_n^{ij})$ be defined for as follows ($i \in \{1, ..., n\}$):

$$a_l^{ij} = \begin{cases} \frac{1}{|r_i \circ r_j|} & \text{if } r_l \in r_i \circ r_j \\ 0 & \text{otherwise} \end{cases}$$

where we define $id \circ r_j = r_j$ for every $j \in \{1, ..., n\}$. Note that $\mathbf{a}_{ij}$ indeed satisfies the requirements of the model: the coordinates of $\mathbf{a}_{ij}$ are non-negative and sum to 1, while $\mathbf{a}_{1j} = one\text{-}hot(j)$ follows from the fact that $id \circ r_j = r_j$. Furthermore, we define $\mathbf{r}_j = one\text{-}hot(j)$.

We show the result by induction. For $i = 0$, if $\mathcal{F}$ contains a fact of the form $r_l(h, e)$, we have $X_e^{(0)} = \{r_l\}$. We show that $\mathbf{e}^{(1)}$ is non-zero in the $l^{th}$ coordinate and zero everywhere else. We have:

$$\phi(\mathbf{h}^{(0)}, \mathbf{r}_l) = \sum_i \sum_l h_i^0 r_j^l \mathbf{a}_{ij} = \sum_j r_j^l \mathbf{a}_{1j} = \mathbf{r}_l = one\text{-}hot(l)$$

This already shows that $\mathbf{e}^{(1)}$ is zero in all coordinates apart from the $l^{th}$. Now let $f \neq h$ and $r_p$ be such that $r_p(f, e) \in \mathcal{F}$. We have

$$\phi(\mathbf{f}^{(0)}, \mathbf{r}_p) = \sum_i \sum_j f_i^0 r_j^p \mathbf{a}_{ij} = \frac{1}{n} \sum_i \sum_j r_j^p \mathbf{a}_{ij} = \frac{1}{n} \sum_i \mathbf{a}_{ip}$$

We need to show that the $l^{th}$ coordinate of the latter vector is non-zero. Since $\mathcal{K} \cup \mathcal{F}$ is consistent and $\mathcal{F}$ contains both $r_l(h, e)$ and $r_p(f, e)$, it has to be the case there there is some $q \in \{1, ..., n\}$ such that $r_l \in r_q \circ r_p$. There thus exists some $q \in \{1, ..., n\}$ such that the $l^{th}$ coordinate of $\mathbf{a}_{qp}$ is non-zero. Since all vectors of the $\mathbf{a}_{ip}$ vectors have non-negative coordinates, it follows that the $l^{th}$ coordinate of $\frac{1}{n} \sum_i \mathbf{a}_{ip}$ is non-zero. We have thus shown that $\{r_j \mid e_j^0 > 0, 2 \leq j \leq n\} = \{r_l\} = X_e^{(0)} \supseteq X_e^{(1)}$.

If $\mathcal{F}$ does not contain any facts of the form $r_l(h, e)$, then $X_e^{(0)} = \mathcal{R}$. We need to show for each $l \in \{2, ..., n\}$ that either $e_l^1$ is non-zero or $r_l \notin X_e^1$. We have that $e_l^1$ is non-zero if the $l^{th}$ component of $\phi(\mathbf{f}^{(0)}, \mathbf{r}_p)$ is non-zero for every fact of the form $r_p(f, e)$ in $\mathcal{F}$, where $f \neq h$. Consider such a fact $r_p(f, e)$ and assume the $l^{th}$ component of $\phi(\mathbf{f}^{(0)}, \mathbf{r}_p)$ is actually 0. Since $f \neq h$ we have $\mathbf{f}^{(0)} = (\frac{1}{n}, ..., \frac{1}{n})$. The $l^{th}$ component of $\phi(\mathbf{f}^{(0)}, \mathbf{r}_p) = \frac{1}{n} \sum_i \mathbf{a}_{ip}$ can thus only be 0 if the $l^{th}$

component of $\mathbf{a}_{ip}$ is 0 for every $i$. This implies that $r_l \notin r_i \circ r_p$ for any $i \in \{1, ..., n\}$, from which it follows that $r_l \notin X_e^{(1)}$.

Now suppose that $X_e^{(i-1)} \supseteq \{r_j \mid e_j^i > 0, 2 \leq j \leq n\} \supseteq X_e^{(i)}$ holds for every entity $e$. We show that (13) must then hold as well. We first show $X_e^{(i)} \supseteq \{r_j \mid e_j^{i+1} > 0, 2 \leq j \leq n\}$. To this end, we need to show that for each $r_j \in X_e^{(i-1)} \setminus X_e^{(i)}$ it holds that $e_j^{i+1} = 0$. If $r_j \in X_e^{(i-1)} \setminus X_e^{(i)}$ it means that there is some $r_p(f, e) \in \mathcal{F}$ such that $r_j \notin X_f^{(i-1)} \diamond r_p$. This is the case if $r_j \notin r_q \diamond r_p$ for every $r_q \in X_f^{(i-1)}$. This, in turn, means that the $j^{\text{th}}$ component of $\mathbf{a}_{qp}$ is 0 for every $q$ such that $r_q \in X_f^{(i-1)}$. Furthermore, by the induction hypothesis, we know that $r_q \notin X_f^{(i-1)}$ means $f_q^i = 0$. In other words, for each $q$ we have that either $f_q^i = 0$ or that the $j^{\text{th}}$ component of $\mathbf{a}_{qp}$ is 0. It follows that the $j^{\text{th}}$ component of $\phi(\mathbf{f}^{(i)}, \mathbf{r}_p)$ is 0, and thus also that $e_j^{i+1} = 0$.

We now show $\{r_j \mid e_j^{i+1} > 0, 2 \leq j \leq n\} \supseteq X_e^{(i+1)}$. Let $j$ be such that $e_j^i > e_j^{i+1} = 0$. We need to show that $r_j \notin X_e^{(i+1)}$. If $e_j^i > e_j^{i+1} = 0$ there needs to be some $r_p(f, e) \in \mathcal{F}$ such that the $j^{\text{th}}$ component of $\phi(\mathbf{f}^{(i)}, \mathbf{r}_p)$ is 0. We have

$$\phi(\mathbf{f}^{(i)}, \mathbf{r}_p) = \sum_l \sum_j f_l^i r_j^p \mathbf{a}_{lj} = \sum_l f_l^i \mathbf{a}_{lp}$$

So when the $j^{\text{th}}$ component of $\phi(\mathbf{f}^{(i)}, \mathbf{r}_p)$ is 0 we must have for each $l$ that either $f_l^i = 0$ or that $a_{lp}^j = 0$. By the induction hypothesis, $f_l^i = 0$ implies $r_l \notin X_f^{(i)}$. Furthermore $a_{lp}^j = 0$ means that $r_j \notin r_l \circ r_p$. When the $j^{\text{th}}$ component of $\phi(\mathbf{f}^{(i)}, \mathbf{r}_p)$ is 0, we thus have that either $r_l \notin X_f^{(i)}$ or $r_j \notin r_l \circ r_p$ for each $l$, which implies $r_j \notin X_e^{(i+1)}$. $\qquad\square$

Another possibility is to use the component-wise product for pooling embeddings:

$$\psi_{\odot} = \frac{(\mathbf{x_1} \odot \ldots \odot \mathbf{x_k})}{\|(\mathbf{x_1} \odot \ldots \odot \mathbf{x_k})\|_1}$$

where we write $\odot$ for the Hadamard product. Using the same argument as in the proof of Proposition 4, we can show that the same result holds for this pooling operator. In practice, however, for numerical stability, we evaluate $\psi_{\odot}$ as follows:

$$\psi_{\odot}(\mathbf{x_1}, \ldots, \mathbf{x_k}) = \frac{(\mathbf{x_1} \odot \ldots \odot \mathbf{x_k}) + \mathbf{z}}{\|(\mathbf{x_1} \odot \ldots \odot \mathbf{x_k}) + \mathbf{z}\|_1} \tag{14}$$

where $\mathbf{z} = (\varepsilon, \ldots, \varepsilon)$ for some small constant $\varepsilon > 0$. As long as $\varepsilon$ is sufficiently small, this does not affect the ability of the GNN model to simulate the directional closure algorithm. In particular, whenever $e_j^i = 0$ in the GNN with $\psi_{\min}$ we can ensure that this coordinate is arbitrarily small in the GNN with $\Psi_{\odot}$ (choosing the $\mathbf{a}_{ij}$ and $\mathbf{r}_j$ vectors as in the proof of Proposition 4), by selecting $\varepsilon$ small enough. In particular, there exists some $\delta > 0$ such that

$$X_e^{(i)} \supseteq \{r_j \mid e_j^{i+1} > \delta, 2 \leq j \leq n\} \supseteq X_e^{(i+1)}$$

# E  DISJUNCTIVE REASONING BENCHMARKS

We introduce two new benchmarks: one based on RCC-8 and one based in IA. Both benchmarks are similar to CLUTRR and Graphlog in their focus on assessing inductive relational reasoning via relation classification queries of the form $(h, ?, t)$, where we need to predict which relation holds between a given head entity $h$ and tail entity $t$. The available knowledge is provided as a set of facts $\mathcal{F}$, which we can again think of as a graph. The inductive part refers to the fact that the model is trained and tested on distinct graphs. The main difficulty comes from the fact that only small graphs are available for training, while some of the test graphs are considerably larger.

However, different from existing benchmarks, our benchmarks require reasoning about disjunctive rules, which is particularly challenging for many methods. The kind of knowledge that has to be

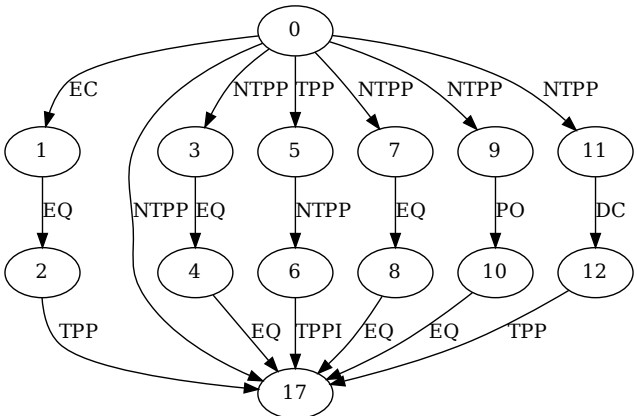

Figure 6: An example from the RCC-8 benchmark with $b = 6$ paths from the source node 0 to the target node 17. Each path has a length of $k = 3$ where $k$ is the number of hops or edges from the node 0 to node 17. The graph has to be collapsed into the single relation $\mathsf{ntpp}(0, 17)$ using information from all the paths.

learned is thus more expressive than the Horn rules which are considered in most existing benchmarks. As illustrated in Example 2, this means that models need to process different relational paths and aggregate the resulting information. We vary the difficulty of problem instances based on the number $b$ of such paths and their length $k$. Figure 6 provides an example where there are $b = 6$ paths of length $k = 3$. Each path is partially informative, in the sense that it allows to exclude certain candidate relations, but is not sufficiently informative to pinpoint the exact relation. The model thus needs to rely on the different paths to be able to exclude all but one of the eight possible relations.

In summary, within the context of benchmarks for systematic generalization, our RCC-8 and IA benchmarks are novel on two fronts compared to existing benchmarks such as CLUTRR (Sinha et al., 2019) and Graphlog (Sinha et al., 2020):

1. **Going beyond Horn rules for relation composition**: The composition of some RCC-8 relations is a disjunction of several RCC-8 relations, and similar for IA.

2. **Multi-path information aggregation**: Models need to reason about multiple relational paths (for the case where $b > 1$) to infer the correct answer.

3. **Rich graph topologies**: The graph generation process recursively expands an edge in the base graph with valid subgraphs and leads to significantly diverse graph topolgies

### E.1   RCC-8

Region Connection Calculus (RCC-8) (Randell et al., 1992) uses eight primitive relations to describe qualitative spatial relationships between regions: $\mathsf{ntpp}(a, b)$ means that $a$ is a proper part of the interior of $b$, $\mathsf{tpp}(a, b)$ means that $a$ is a proper part of $b$ and shares a boundary point with $b$, $\mathsf{po}(a, b)$ means that $a$ and $b$ are overlapping (but neither is included in the other), $\mathsf{dc}(a, b)$ means that $a$ and $b$ are disjoint, $\mathsf{ec}(a, b)$ means that $a$ and $b$ are adjacent (i.e. sharing a boundary point but no interior points), $\mathsf{eq}(a, b)$ means that $a$ and $b$ are equal, and $\mathsf{ntppi}$ and $\mathsf{tppi}$ are the inverses of $\mathsf{ntpp}$ and $\mathsf{tpp}$.

**Example 2.** *The RCC-8 calculus describes qualitative spatial relations between two regions using eight primitive relations. The semantics of the RCC-8 relations are governed by disjunctive rules such as:*

$$\mathsf{po}(X, Z) \vee \mathsf{tpp}(X, Z) \vee \mathsf{ntpp}(X, Z) \leftarrow \mathsf{ec}(X, Y) \wedge \mathsf{ntpp}(Y, Z) \tag{15}$$

$$\mathsf{po}(X, Z) \vee \mathsf{tppi}(X, Z) \vee \mathsf{ntppi}(X, Z) \leftarrow \mathsf{tppi}(X, Y) \wedge \mathsf{po}(Y, Z) \tag{16}$$

Table 5: RCC-8 composition table (Cui et al., 1993) (excluding eq).

| | | dc | ec | po | tpp | ntpp | tppi | ntppi |
|---|---|---|---|---|---|---|---|---|
| dc | | $\mathcal{R}_8$ | dc, ec, po, tpp, ntpp | dc, ec, po, tpp, ntpp | dc, ec, po, tpp, ntpp | dc, ec, po, tpp, ntpp | dc | dc |
| ec | | dc, ec, po, tppi, ntppi | dc, ec, po, tpp, tppi, eq | dc, ec, po, tpp, ntpp | ec, po, tpp, ntpp | po, tpp, ntpp | dc, ec | dc |
| po | | dc, ec, po, tppi, ntppi | dc, ec, po, tppi, ntppi | $\mathcal{R}_8$ | po, tpp, ntpp | po, tpp, ntpp | dc, ec, po, tppi, ntppi | dc, ec, po, tppi, ntppi |
| tpp | | dc | dc, ec | dc, ec, po, tpp, ntpp | tpp, ntpp | ntpp | dc, ec, po, tpp, tppi, eq | dc, ec, po, tppi, ntppi |
| ntpp | | dc | dc | dc, ec, po, tpp, ntpp | ntpp | ntpp | dc, ec, po, tpp, ntpp | $\mathcal{R}_8$ |
| tppi | | dc, ec, po, tppi, ntppi | ec, po, tppi, ntppi | po, tppi, ntppi | po, eq, tpp, tppi | po, tpp, ntpp | tppi, ntppi | ntppi |
| ntppi | | dc, ec, po, tppi, ntppi | po, tppi, ntppi | po, tppi, ntppi | po, tppi, ntppi | po, tppi, tpp, ntpp, ntppi, eq | ntppi | ntppi |

*as well as constraints encoding that the RCC-8 relations are disjoint. Let $\mathcal{K}$ contain these rules and constraints, and let $\mathcal{F} = \{\text{ec}(a, b), \text{ntpp}(b, c), \text{tppi}(a, d), \text{po}(d, c)\}$. Then we have $\mathcal{K} \cup \mathcal{F} \models \text{po}(a, c)$. Indeed, using (15) we can infer $\text{po}(a, c) \lor \text{tpp}(a, c) \lor \text{ntpp}(a, c)$, while using (16) we can infer $\text{po}(a, c) \lor \text{tppi}(a, c) \lor \text{ntppi}(a, c)$. Using the disjointness constraints, we finally infer $\text{po}(a, c)$.*

The RCC-8 semantics is governed by the so-called composition table, which describes the composition mapping between two relations. This is shown in Table 5 where the trivial composition with the identity element eq being itself is dropped. Each entry in this table corresponds to a rule of the form (15), specifying the possible relations that may hold between two regions $a$ and $c$, when we know the RCC-8 relation that holds between $a$ and some region $b$ as well as the relation that holds between $b$ and $c$. For instance, the composition of ec and tppi is given by the set $\{\text{dc}, \text{ec}\}$, which means that from $\{\text{ec}(a, b), \text{tppi}(b, c)\}$ we can infer $\text{dc}(a, c) \lor \text{ec}(a, c)$. In the table, we write $\mathcal{R}_8$ to denote that any RCC-8 relation is possible.

### E.2    ALLEN INTERVAL ALGEBRA

We also introduce a benchmark based on Allen's interval algebra (Allen, 1983b) for qualitative temporal reasoning. The interval algebra (IA) uses 13 primitive relations to describe qualitative temporal relationships. IA captures all possible relationships between two time intervals, as follows: $<(a, b)$ means that the time interval $a$ completely precedes the time interval $b$; $\text{d}(a, b)$ means that $a$ occurs during $b$ while not sharing any boundary points; $\text{o}(a, b)$ means that $a$ overlaps with $b$; $\text{m}(a, b)$ means that $a$ meets $b$ (i.e. $a$ ends exactly when $b$ starts); $\text{s}(a, b)$ means that $a$ starts $b$ (i.e. $a$ and $b$ start at the same time while $a$ finishes strictly before $b$); $\text{f}(a, b)$ means that $a$ finishes $b$ ($a$ and $b$ finish at the same time, while $b$ starts strictly before $a$); $=(a, b)$ means that $a$ equals $b$; and finally $>, \text{di}, \text{oi}, \text{mi}, \text{si}, \text{fi}$ are the inverses of the respective operations defined previously. The composition table for all the primitive interval relations is shown in Table 6 with the exception of the trivial composition of primitive elements with the identity element $=$.

### E.3    DATASET GENERATION PROCESS

We now explain how the dataset was created. All sampling in the discussion below is uniform random. Each problem instance has to be constructed such that after aggregating the information provided by all the relational paths, we need to be able to infer a singleton label. In other words, problem instances need to be consistent (i.e. the information provided by different paths cannot be conflicting) and together all the paths need to be informative enough to uniquely determine which relation holds between the head and tail entity. This makes brute-force sampling of problem instances prohibitive. Instead, to create a problem instance involving $b$ paths of length $k$, we first sample a base graph, which has $b$ shorter paths, with a length in $\{2, 3, 4\}$. This is done by pre-computing relational compositions for a large number of paths and then selecting $b$ paths whose intersection is a singleton. Then we repeatedly increase the length of the paths by selecting an edge and replacing it by a short path whose composition is equal to the corresponding relation.

Finally, to add further diversity to the graph topology, for each of the $b$ paths, we allow 1 edge from the base graph to be replaced by a subgraph (rather than a path), where this subgraph is generated

Table 6: Allen's interval algebra composition table (Allen, 1983b) excluding the trivial composition with =.

| | < | > | d | di | o | oi | m | mi | s | si | f | fi |
|---|---|---|---|---|---|---|---|---|---|---|---|---|
| < | < | | <, o, m, d, s | < | < | <, o, m, d, s | < | <, o, m, d, s | < | < | <, o, m, d, s | < |
| > | | > | >, oi, mi, d, f | > | >, oi, mi, d, f | > | >, oi, mi, d, f | > | >, oi, mi, d, f | > | > | > |
| d | < | > | d | | <, o, m, d, s | >, oi, mi, d, f | < | > | d | >, oi, mi, d, f | d | <, o, m, d, s |
| di | <, o, m, di, fi | >, oi, di, mi, si | o, oi, d, s, f, di, si, fi, = | di | o, di, fi | oi, di, si | o, di, fi | oi, di, si | o, di, fi | di | oi, di, si | di |
| o | < | >, oi, di, mi, si | o, d, s | <, o, m, di, fi | <, o, m | o, oi, d, s, f, di, si, fi, = | < | oi, di, si | o | o, di, fi | o, d, s | <, o, m |
| oi | <, o, m, di, fi | > | oi, d, f | >, oi, mi, di, si | o, oi, d, di, s, si, f, fi, = | >, oi, mi | o, di, fi | > | oi, d, f | oi, >, mi | oi | oi, di, si |
| m | < | >, oi, di, mi, si | o, d, s | < | < | o, d, s | < | f, fi, = | m | m | d, s, o | < |
| mi | <, o, m, di, fi | > | oi, d, f | > | oi, d, f | > | s, si, = | > | d, f, oi | > | mi | mi |
| s | < | > | d | <, o, m, di, fi | <, o, m | oi, d, f | < | mi | s | s, si, = | d | <, m, o |
| si | <, o, m, di, fi | > | oi, d, f | di | o, di, fi | oi | o, di, fi | mi | s, si, = | si | oi | di |
| f | < | > | d | >, oi, mi, di, si | o, d, s | >, oi, mi | m | > | d | >, oi, mi | f | f, fi, = |
| fi | < | >, oi, di, mi, si | o, d, s | di | o | oi, di, si | m | si, oi, di | o | di | f, fi, = | fi |

using the same procedure. Note that the final path count $b$ then includes the paths from this subgraph as well.

The process is described in more detail below:

1. **Sample short paths:** Randomly sample $n = 100\,000$ paths of length $k \in \{2, 3, 4\}$ and compute their composition. Note that this sampling is done with replacement to avoid uniqueness upper bounds for small graphs.

2. **Generate base graphs:** Generate the desired number of $b$-path base graphs, by selecting paths that were generated in step 1. Each individual path typically composes to a set of relations, but the graphs are constructed such that the intersection of these sets, across all $b$ paths, produces a singleton target label.

3. **Recursive edge expansion:** Randomly pick an edge from a path that does not yet have the required length $k$. Select a path from step 1 which composes to a singleton, corresponding to the relation that is associated with the chosen edge. Replace the edge with this path.

4. **Recursive subgraph expansion:** Rather than replacing an edge with a path, we can also replace it with a subgraph. As candidate subgraphs, we use the base graphs from step 2 with at most $\lfloor \frac{b}{2} \rfloor$ paths.

5. Keep repeating steps 2 and 3 until we have the desired number paths $b$ with the desired length of $k$, with the restriction that step 3 is applied at most once to each path from the initial base graph.

Some example graphs generated via this procedure for the RCC-8 dataset are displayed in Figure 7. For higher $k$, there is greater diversity in the graph topology and complexity of the graph. To create a

dataset for reasoning about interval algebra problems, we follow the same process as for the RCC-8 dataset. Example graphs generated via this procedure for IA are displayed in Figure 8.

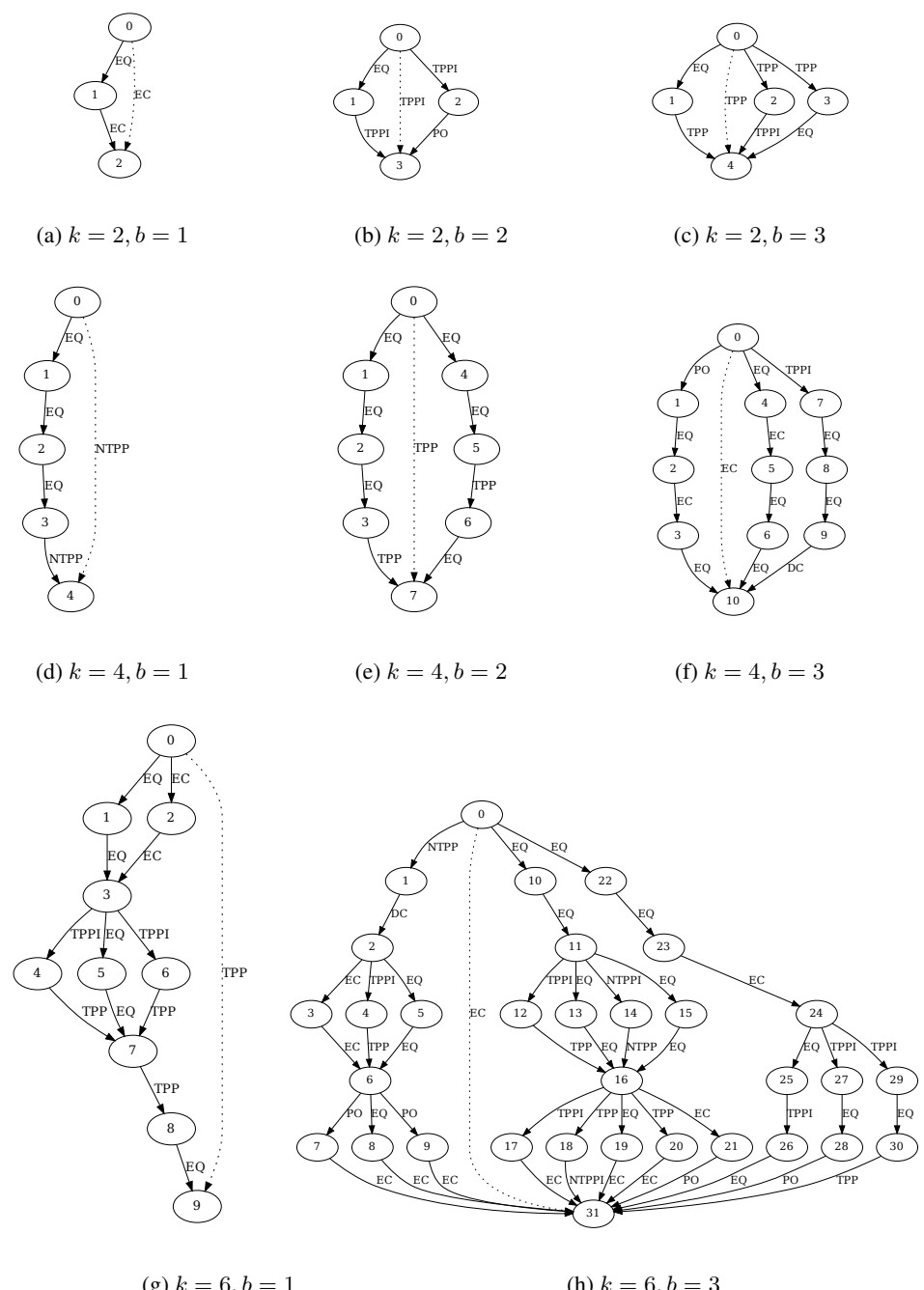

Figure 7: Some graph instances for the RCC-8 dataset generated using the procedure described in E.3. The graph topology becomes more diverse for the test instances when sub-graphs are embedded within a single path, as shown in (g) for path length $k = 6$ and number of paths $b = 1$. In this particular case, there are two sub-graphs that have been embedded in the graph by replacing two edges. Instances of the type shown in (a), (b), (c), (d), (e), (f) are used in the training set and the graph topology is fixed in this case. The target edge label between the source node and the tail node that needs to be predicted by the model is indicated by the dotted line.

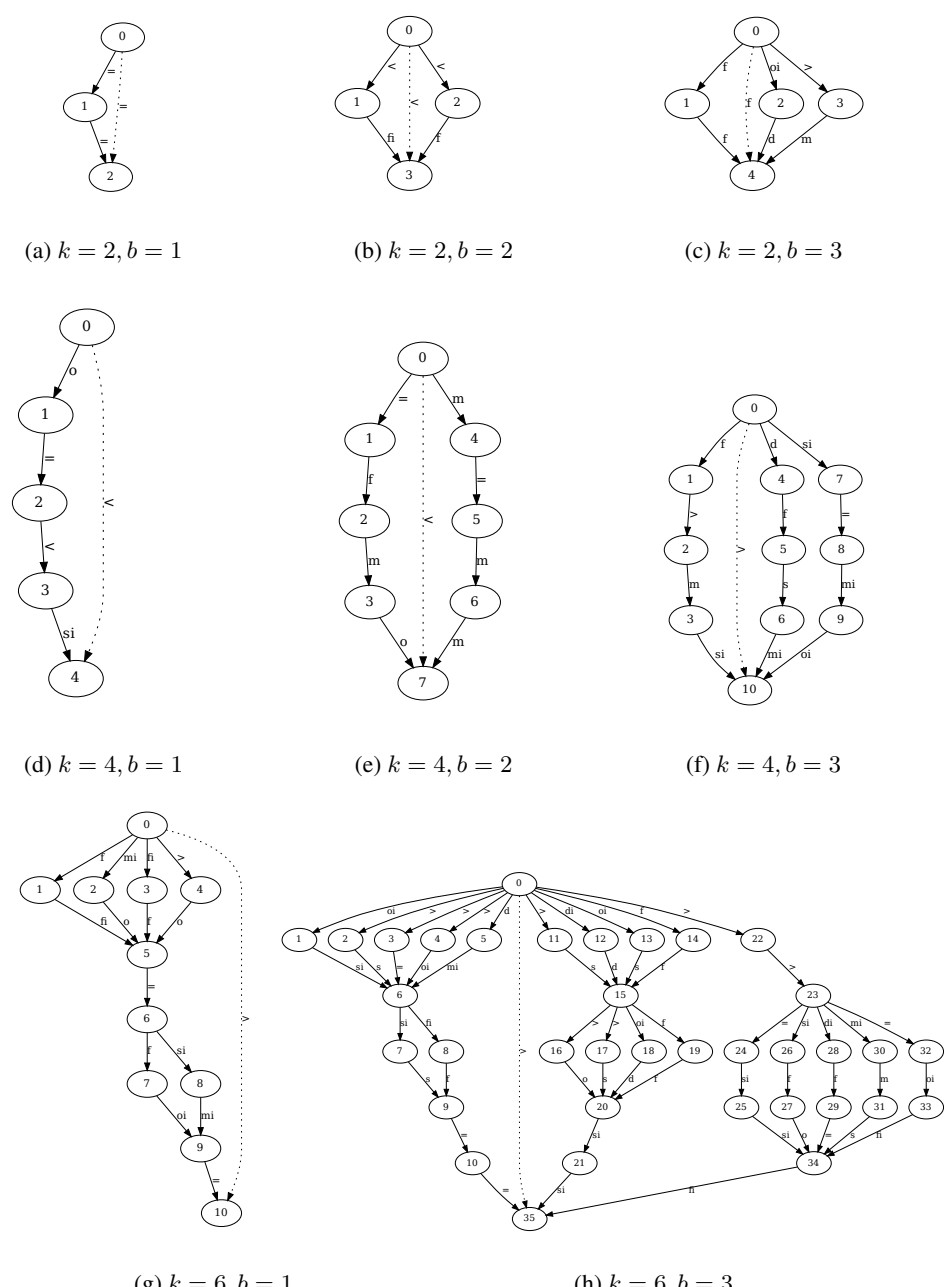

Figure 8: Graph instances for the interval dataset generated using the procedure described in E.3. We highlight the rich variation in the topology of the graph instance for path length $k = 6$ and number of paths $b = 3$ in (h) by contrasting it with a similar graph instance for the RCC-8 dataset shown in Figure 7(h). The target edge label between the source node and the tail node that needs to be predicted by the model is indicated by the dotted line.

## E.4 ENSURING PATH CONSISTENCY WITHIN THE DATASET

We ensure that all relational paths in a problem instance in the generated dataset do not informationally conflict with each other by using the DPC+ algorithm (Long et al., 2016). It efficiently computes directional path consistency, i.e. $X_{ij} \subseteq X_{ik} \diamond X_{kj} \forall i, j \leq k$, for qualitative constraint networks that we can transform our graph instances to. Note that this takes advantages of the fact that directional

path consistency is sufficient as a test for global path consistency for networks with singleton edge labels (Li et al., 2015).

# F    ADDITIONAL EXPERIMENTAL DETAILS

We now present further details of the experimental set-up, including details of the loss function that was used for training the model, the considered benchmarks and baselines, and training details such as hyperparameter optimisation.

## F.1    LOSS FUNCTION

**Forward model**    We first consider the base setting, where a single forward model is used. Let a set of training instances of the form $(\mathcal{F}_i, h_i, t_i, r_i)$ be given, where we write $\mathbf{t}_i = (t_{i,1}, \ldots, t_{i,n})$ for the final-layer embedding of entity $t_i$ in the graph associated with $\mathcal{F}_i$. Let $\mathbf{r} = (r_1, \ldots, r_n)$ denote the embedding of relation $r$. We write:

$$\mathrm{CE}(\mathbf{t}_i, \mathbf{r}) = -\sum_{j=1}^{n} r_j \log t_{i,j} \tag{17}$$

Since $r_i$ represents the correct label for training instance $i$, we clearly want $\mathrm{CE}(\mathbf{t}_i, \mathbf{r_i})$ to be low, while for each negative example $r' \in \mathcal{R} \setminus \{r_i\}$ we want $\mathrm{CE}(\mathbf{t}_i, \mathbf{r}')$ to be high. We implement this with a margin loss, where for each $i$ we let $r_i' \in \mathcal{R} \setminus \{r_i\}$ be a corresponding negative example:

$$\mathcal{L} = \sum_{i} \max(0, \mathrm{CE}(\mathbf{t}_i, \mathbf{r_i}) - \mathrm{CE}(\mathbf{t}_i, \mathbf{r}_i') + \Delta) \tag{18}$$

where the margin $\Delta > 0$ is a hyperparameter.

**Full model**    In general, we use $m$ different models, each intuitively capturing a different aspect of the relations. Furthermore, instead of the tail node embedding $\mathbf{t_i}$, we use the prediction obtained by the forward-backward model, computed as in (5). Let us write $\mathbf{x_{ij}}$ to denote the prediction that is obtained by the $j^{\text{th}}$ model for training example $i$, and let $\mathbf{r_{ij}}$ denote the embedding of relation $r_i$ in the $j^{\text{th}}$ model. We write $\mathbf{r}'_{\mathbf{ij}}$ to denote some negative example, i.e. $\mathbf{r}'_{\mathbf{ij}} = \mathbf{r_{pj}}$ for some $r_p \in \mathcal{R} \setminus \{r_i\}$. The overall loss function becomes:

$$\mathcal{L} = \sum_{i} \max\left(0, \left(\sum_{j=1}^{m} \mathrm{CE}(\mathbf{x_{ij}}, \mathbf{r_{ij}}) - \mathrm{CE}(\mathbf{x_{ij}}, \mathbf{r}'_{\mathbf{ij}})\right) + \Delta\right) \tag{19}$$

## F.2    INFERENCE

**Relation classification**    For the relation classification datasets, where we need to answer queries of type $(h, ?, t)$, at test time, we predict the target relation for which the cross-entropy with the predicted embedding is minimal. More precisely, let us write $\mathbf{x_j}$ for the embedding predicted by the $j^{\text{th}}$ model, computed as in (5). Let $\mathbf{r_j}$ be the embedding of relation $r$ in the $j^{\text{th}}$ model. We predict the relation $\hat{r}$ defined as follows

$$\hat{r} = \arg\max_{r \in \mathcal{R}} \sum_{j=1}^{m} \mathrm{CE}(\mathbf{x_j}, \mathbf{r_j}) \tag{20}$$

**Link prediction**    For link prediction datasets, where we need to answer queries of type $(h, r, ?)$, we use the forward-only model, as we need to efficiently compute a score for all the entities in the knowledge graph. This is possible with one pass of the forward model, whereas with the full (forward-backward) model we would have to do one pass for each candidate entity. Let $\mathbf{t_j}$ be the final-layer embedding of entity $t$ in the $j^{\text{th}}$ model. Let us again write $\mathbf{r_j}$ for the embedding of relation $r$ in the $j^{\text{th}}$ model. The score of entity $t$ is then given by:

$$score(t) = \sum_{j=1}^{m} \mathrm{CE}(\mathbf{t_j}, \mathbf{r_j}) \tag{21}$$

Finally, to answer the link prediction query, we rank the set of all candidate entities based on this score.

Table 7: Data statistics for different versions of the CLUTRR dataset, with varying training regimes and different numbers training and testing graphs.

| Training regime | Unique Hash | No. of relations | # Train | # Test | Test regime |
|---|---|---|---|---|---|
| $k \in \{2, 3\}$ | data_089907f8 | 22 | 10,094 | 900 | $k \in \{4, \ldots, 10\}$ |
| $k \in \{2, 3\}$ | data_9b2173cf | 22 | 35,394 | 39825 | $k \in \{4, \ldots, 10\}$ |
| $k \in \{2, 3, 4\}$ | data_db9b8f04 | 22 | 15,083 | 823 | $k \in \{5, \ldots, 10\}$ |

Table 8: Data statistics of the Spatio-temporal reasoning datasets

| Dataset | Training regime | No. of relations | # Train | # Test | Test regime |
|---|---|---|---|---|---|
| RCC-8 | $b \in \{1, 2, 3\}, k \in \{2, 3\}$ | 8 | 57,600 | 153,600 | $b \in \{1, 2, 3\}, k \in \{2, \ldots, 9\}$ |
| IA | $b \in \{1, 2, 3\}, k \in \{2, 3\}$ | 13 | 57,600 | 153,600 | $b \in \{1, 2, 3\}, k \in \{2, \ldots, 9\}$ |

## F.3 BENCHMARKS

**CLUTRR**[1] (Sinha et al., 2019) is a dataset which involves reasoning about family relationships. The original version of the dataset involved narratives describing the fact graph in natural language. It was, among others, aimed at testing the ability of language models such as BERT (Devlin et al., 2019) to solve such reasoning tasks. However, the original paper also considered a number of baselines which were given access to the fact graph itself, especially GNNs and sequence classification models. A crucial finding was that such models fail to learn to reason in a systematic way: models trained on short inference chains perform poorly when tested on examples involving longer inference chains. This has inspired a line of work which has introduced a number of neuro-symbolic methods for addressing this issue. The CLUTRR dataset was released under a CC-BY-NC 4.0 license.

**Graphlog**[2] (Sinha et al., 2020) involves examples for 57 different *worlds*, where each world is characterised by a set of logical rules. For each world, a number of corresponding knowledge graphs are provided, which the model can use to learn the underlying rules. The model is then tested on previously unseen knowledge graphs for the same world. The aim of this benchmark is to test the ability of models to systematically generalise from the reasoning patterns that have been observed during training, i.e. to apply the rules that have been learned from the training data in novel ways. This dataset is released under a CC-BY-NC 4.0 license.

**RCC-8** and **IA** are the benchmarks that we introduce in this paper, as described in Section E. We release these benchmarks under a CC-BY 4.0 license.

**Inductive Knowledge Graph Completion**[3] (Teru et al., 2020) is focused on link prediction queries of the form $(h, r, ?)$ which are evaluated against a given knowledge graph. Different from the more commonly used *transductive* setting, in the case of *inductive* knowledge graph completion, the training and test knowledge graphs are disjoint. Teru et al. (2020) proposed a number of benchmarks for this inductive setting by sampling disjoint training and test graphs from standard knowledge graph completion datasets. In this way, they obtained four different benchmarks from FB15k-237 and four benchmarks from WN18RR.

Dataset statistics for CLUTRR, Graphlog, RCC-8 and IA, and the Inductive KGC benchmarks are reported in tables 7, 9, 8, 10. We use a standard 80-20 split for training and validation for CLUTRR and RCC-8. For Graphlog, we use the validation set that is provided separately from the test set.

## F.4 BASELINES

We compare our method against the following neuro-symbolic methods:

**CTP** Conditional Theorem Provers (Minervini et al., 2020b) are a more efficient version of Neural Theorem Provers (NTPs (Rocktäschel & Riedel, 2017)). Like NTPs, they learn a differ-

---

[1] https://github.com/facebookresearch/clutrr
[2] https://github.com/facebookresearch/graphlog
[3] https://github.com/kkteru/grail

Table 9: Data statistics for the 'hard' Graphlog worlds, showing for each world the number of classes (NC), the number of distinct resolution sequences (ND), the average resolution length (ARL), the average number of nodes (AN), the average number of edges (AE), and the number of training and testing graphs.

| World ID | NC | ND | ARL | AN | AE | # Train | #Test |
|---|---|---|---|---|---|---|---|
| World 6 | 16 | 249 | 5.06 | 16.3 | 20.2 | 5000 | 1000 |
| World 7 | 17 | 288 | 4.47 | 13.2 | 16.3 | 5000 | 1000 |
| World 8 | 15 | 404 | 5.43 | 16.0 | 19.1 | 5000 | 1000 |
| World 11 | 17 | 194 | 4.29 | 11.5 | 13.0 | 5000 | 1000 |
| World 32 | 16 | 287 | 4.66 | 16.3 | 20.9 | 5000 | 1000 |

Table 10: Dataset statistics for inductive knowledge graph completion. Queries and facts are $(h, r, t)$ triplets and are used as *labels* and *inputs* respectively. The goal is to predict the query targets $t$ once trained on fact triplets. Note that for the training sets, queries are treated as facts i.e. training data.

| Dataset | | #Relation | Train | | | Validation | | | Test | | |
|---|---|---|---|---|---|---|---|---|---|---|---|
| | | | #Entity | #Query | #Fact | #Entity | #Query | #Fact | #Entity | #Query | #Fact |
| FB15k-237 | v1 | 180 | 1594 | 4245 | 4245 | 1594 | 489 | 4245 | 1093 | 205 | 1993 |
| | v2 | 200 | 2608 | 9739 | 9739 | 2608 | 1166 | 9739 | 1660 | 478 | 4145 |
| | v3 | 215 | 3668 | 17986 | 17986 | 3668 | 2194 | 17986 | 2501 | 865 | 7406 |
| | v4 | 219 | 4707 | 27203 | 27203 | 4707 | 3352 | 27203 | 3051 | 1424 | 11714 |
| WN18RR | v1 | 9 | 2746 | 5410 | 5410 | 2746 | 630 | 5410 | 922 | 188 | 1618 |
| | v2 | 10 | 6954 | 15262 | 15262 | 6954 | 1838 | 15262 | 2757 | 441 | 4011 |
| | v3 | 11 | 12078 | 25901 | 25901 | 12078 | 3097 | 25901 | 5084 | 605 | 6327 |
| | v4 | 9 | 3861 | 7940 | 7940 | 3861 | 934 | 7940 | 7084 | 1429 | 12334 |

entiable logic program, but rather than exhaustively considering all derivations, at each step of a proof, CTPs learn a filter function that selects the most promising rules to apply, thereby speeding up backwards-chaining procedure of NTP. Three variants of this model were proposed, which differ in how this selection step is done, i.e. using a linear mapping ($\text{CTP}_\text{L}$), using an attention mechanism ($\text{CTP}_\text{A}$), and using a method inspired by key-value memory networks (Miller et al., 2016) ($\text{CTP}_\text{M}$). We were not able to reproduce the results from the original paper, hence we report the results from (Minervini et al., 2020b) for the CLUTRR benchmark.

**GNTP** Greedy NTPs (Minervini et al., 2020a) are another approximation of NTPs, which select the top-$k$ best matches during each inference step.

**R5** This model (Lu et al., 2022) learns symbolic rules of the form $r(X, Z) \leftarrow r_1(X, Y) \wedge r_2(Y, Z)$, with the possibility of using invented predicates in the head. To make a prediction, the method then samples (or enumerates) simple paths between the head and tail entities and iteratively applies the learned rules to reduce these paths to a single relation. The order in which relations are composed is determined by Monte Carlo Tree Search.

**NCRL** Neural Compositional Rule Learning (Cheng et al., 2023) also samples relational paths between the head and tail entities, and iteratively reduces them by composing 2 relations at a time, similar to R5. However, in this case, the choice of the two relations to compose in each step are determined by a Recurrent Neural Network. Moreover, rather than learning symbolic rules, the rules are learned implicitly by using an attention mechanism to compose relations. Both R5 and NCRL implicitly make the assumption that the relational reasoning problem is about predicting the target relation from a single relational path, and that this prediction can be done by repeatedly applying Horn rules.

The following transformer (Vaswani et al., 2017) variant is also a natural baseline:

**ET** Edge Transformers (Bergen et al., 2021) modify the transformer architecture by using an attention mechanism that is designed to simulate relational composition. In particular, the embeddings are interpreted as representations of edges in a graph. To update the representation of an edge $(a, c)$ the model selects pairs of edges $(a, x), (x, b)$ and composes their embeddings. These compositions are aggregated using an attention mechanism, similar as in the standard transformer architecture.

We also compare against several GNN models:

**GCN**  Graph Convolutional Networks (Kipf & Welling, 2017) are a standard graph neural network architecture, which use sum pooling and rely on a linear layer followed by a non-linearity such as ReLU or sigmoid to compute messages. While standard GCNs do not take into account edge types, for the experiments we concatenate edge types to node embeddings during message passing, following (Sinha et al., 2019). GCNs learn node embeddings and can thus not directly be used for relation classification. To make the final prediction, we combine the final-layer embeddings of the head and tail entities with an encoding of the target relation, and make the final prediction using a softmax classification layer.

**R-GCN**  Relational GCNs (Schlichtkrull et al., 2018) are a variant of GCNs in which messages are computed using a relation-specific linear transformation. This is similar in spirit to how we compute messages in our framework, but without the inductive bias that comes from treating embeddings as probability distributions over primitive relation types.

**GAT**  Graph Attention Networks (Velickovic et al., 2018) are a variant of GCNs, which use a pooling mechanism based on attention. Similar as for GCNs, we concatenate the edge types to node embeddings to take into account the edge types.

**E-GAT**  Edge-based Graph Attention Networks (Sinha et al., 2020) are a variant of GATs which take edge types into account. In particular, an LSTM module is used to combine the embedding of a neighboring node with an embedding of the edge type. The resulting vectors are then aggregated as in the GAT architecture.

**NBFNet**  Neural Bellman-Ford Networks (Zhu et al., 2021) model the relationship between a designated head entity and the other entities from a given graph. Our model employs essentially the same strategy to use GNNs for relation classification, which is to learn entity embeddings that capture the relationship with the head entity rather than the entities themselves. The main difference between NBFnet and our model comes from the additional inductive bias that our model is adding.

In (Minervini et al., 2020b), a number of sequence classifiers were also used as baselines, and we also report these results. These methods sample a path between the head and the tail, encode the path using a recurrent neural network, and then make a prediction with a softmax classification layer. We report results for three types of architectures: vanilla **RNNs**, **LSTMs** (Hochreiter & Schmidhuber, 1997) and **GRUs** (Cho et al., 2014).

## F.5  TRAINING DETAILS

### F.5.1  INITIALIZATION AND COMPUTE

The relation vectors $\mathbf{r}$ and the vectors $\mathbf{a}_{ij}$ defining the composition function are uniformly initialized. All baseline results that were obtained by us were hyperparameter-tuned using grid search, as detailed below. Some baseline results were obtained from their corresponding papers and reported verbatim (as indicated in the results tables). All experiments were conducted using RTX 4090 and V100 NVIDIA GPUs. A single experiment using the GNN based methods in the paper can be conducted within 30 minutes to an hour on a single GPU. This includes training and testing a single model on any benchmark of the following relation prediction benchmarks: CLUTRR, Graphlog, RCC-8, IA (also see Figure 16 for train/test times on the spatiotemporal datasets). A single hyperparameter set evaluation would take the same time as an individual experiment. For the inductive knowledge graph completion setting, training and inference times for a single run are reported in Figure 15.

### F.5.2  HYPERPARAMETER SETTINGS

We use the Adam optimizer (Kingma & Ba, 2017). The number of layers of the EpiGNN model is fixed to 9 and the number of negative examples per instance is fixed as 1. The other hyperparameters of the EpiGNN model are tuned using grid search. The optimal values that were obtained are mentioned in Table 11. For inductive knowledge graph completion, the classification task requires predicting tail entities so we cannot use backward flow in our model. The optimal hyperparameters for the forward only version of the EpiGNN for this setting are summarized in Table 12. For this setting, we use 6 message passing rounds similarly with NBFNet Zhu et al. (2021).

Table 11: Optimal hyperparameters of the full (forward-backward) model on all benchmarks.

| | Batch size | Embedding dim | Epochs | Facets | Learning rate | Margin |
|---|---|---|---|---|---|---|
| CLUTRR | 128 | 64 | 100 | 8 | 0.01 | 1.0 |
| Graphlog | 64 | 64 | 150 | 1 | 0.01 | 1.0 |
| RCC-8 | 128 | 32 | 40 | 4 | 0.01 | 1.0 |
| IA | 128 | 64 | 40 | 8 | 0.01 | 1.0 |

Table 12: Optimal hyperparameters of the forward only EpiGNN model on inductive knowledge graph completion benchmarks.

| | Batch size | Embedding dim | Epochs | Facets | Learning rate | Margin |
|---|---|---|---|---|---|---|
| FB15k-237 v1 | 64 | 64 | 15 | 16 | 0.01 | 0.2 |
| FB15k-237 v2 | 64 | 64 | 15 | 16 | 0.01 | 0.1 |
| FB15k-237 v3 | 64 | 64 | 15 | 16 | 0.01 | 0.1 |
| FB15k-237 v4 | 64 | 64 | 15 | 16 | 0.01 | 0.1 |
| WN18RR v1 | 64 | 16 | 15 | 4 | 0.01 | 0.5 |
| WN18RR v2 | 64 | 16 | 15 | 4 | 0.01 | 0.4 |
| WN18RR v3 | 64 | 64 | 15 | 4 | 0.01 | 1.1 |
| WN18RR v4 | 64 | 16 | 15 | 4 | 0.01 | 0.7 |

We conduct the following hyperparameter sweeps: learning rate in $\{0.00001, 0.001, 0.01, 0.1\}$, batch size in $\{16, 32, 64, 128\}$, number of facets $m$ in $\{1, 2, 4, 8, 16, 32\}$ and embedding dimension size in $\{8, 16, 32, 64, 128, 256\}$. We also tune the margin $\Delta$ in the loss function over $\{10, 1.1, 1.0, 0.9, \ldots, 0.1, 0.01\}$. All model parameters are shared across the different message passing layers of our model.

The choice of the pooling operator has an important impact on the systematic generalization abilities of the model. In our experiments, we found that the pooling operator has to be specified as part of the inductive bias and cannot be learned from the training or validation data, which is in accordance with findings from the literature on systematic generalization (Xu et al., 2021; Bahdanau et al., 2019).

# G ADDITIONAL ANALYSIS

## G.1 ADDITIONAL CLUTRR RESULTS

In the main paper, we presented the results for the standard CLUTRR benchmark, where problems of size $k \in \{2, 3, 4\}$ are used for training. In the literature, models are sometimes also evaluated on an even harder setting, where only problems of size $k \in \{2, 3\}$ are available for training. We show the results for this setting in Table 13. As can be seen, our model clearly outperforms both Edge Transformers (ET) and the GNN and RNN baselines. In this more challenging setting, the difference in performance between our model and ET is much more pronounced. However, R5, as the best-performing neuro-symbolic method, consistently outperforms our method in this case. We hypothesise that this is largely due to the inevitably small size of the training set (as the number of distinct paths of length 3 is necessarily limited). Rule learners can still perform well in such cases, which is something that R5 is able to exploit. To achieve similar results with our model, a stronger inductive bias would have to be imposed. One possibility would be to impose a sparsity prior on the relation embeddings $\mathbf{r}$ and the vectors $\mathbf{a}_{ij}$ defining the composition function. We leave a detailed investigation of this possibility for future work.

In the literature, two different variants of the dataset have been used: `db_9b2173cf` and `data_089907f8`. In Table 13, we use the CLUTRR dataset `db_9b2173cf`, which was introduced in the ET paper (Bergen et al., 2021), to evaluate our model as well as the baselines that were evaluated by us. The reported baseline results that were obtained from (Minervini et al., 2020b) and (Lu et al., 2022) are based on the smaller `data_089907f8` variant, and are thus not directly comparable.

Table 13: Results on CLUTRR (accuracy) after training on problems with $k \in \{2, 3\}$ and then evaluating on problems with $k \in \{4, \ldots, 10\}$. The best performance for each $k$ is highlighted in **bold**. Results marked with $*$ were taken from (Minervini et al., 2020b) and those with $\dagger$ from (Lu et al., 2022). The results from (Minervini et al., 2020b) and (Lu et al., 2022) were evaluated on a different variant of the dataset and may thus not be directly comparable.

| | 4 Hops | 5 Hops | 6 Hops | 7 Hops | 8 Hops | 9 Hops | 10 Hops |
|---|---|---|---|---|---|---|---|
| EpiGNN-mul (ours) | 0.96±.02 | 0.96±.03 | 0.94±.05 | 0.92±.07 | 0.90±.10 | 0.88±.11 | 0.85±.13 |
| EpiGNN-min (ours) | 0.96±.02 | 0.95±.05 | 0.91±.08 | 0.87±.11 | 0.82±.13 | 0.79±.14 | 0.74±.15 |
| $R5^{\dagger}$ | 0.98±.02 | **0.99±.02** | 0.98±.03 | **0.96±.05** | **0.97±.01** | **0.98±.03** | **0.97±.03** |
| $CTP^*_L$ | 0.98±.02 | 0.98±.03 | 0.97±.05 | 0.96±.04 | 0.94±.05 | 0.89±.07 | 0.89±.07 |
| $CTP^*_A$ | **0.99±.02** | 0.99±.01 | **0.99±.02** | 0.96±.04 | 0.94±.05 | 0.89±.08 | 0.90±.07 |
| $CTP^*_M$ | 0.97±.03 | 0.97±.03 | 0.96±.06 | 0.95±.06 | 0.93±.05 | 0.90±.06 | 0.89±.06 |
| $GNTP^*$ | 0.49±.18 | 0.45±.21 | 0.38±.23 | 0.37±.21 | 0.32±.20 | 0.31±.19 | 0.31±.22 |
| ET | 0.90±.04 | 0.84±.02 | 0.78±.02 | 0.69±.03 | 0.63±.05 | 0.58±.06 | 0.55±.08 |
| $GAT^*$ | 0.91±.02 | 0.76±.06 | 0.54±.03 | 0.56±.04 | 0.54±.03 | 0.55±.05 | 0.45±.06 |
| $GCN^*$ | 0.84±.03 | 0.68±.02 | 0.53±.03 | 0.47±.04 | 0.42±.03 | 0.45±.03 | 0.39±.02 |
| NBFNet | 0.55±.08 | 0.44±.07 | 0.39±.07 | 0.37±.06 | 0.34±.04 | 0.32±.05 | 0.31±.05 |
| R-GCN | 0.80±.09 | 0.63±.08 | 0.52±.11 | 0.46±.07 | 0.41±.05 | 0.39±.06 | 0.38±.05 |
| $RNN^*$ | 0.86±.06 | 0.76±.08 | 0.67±.08 | 0.66±.08 | 0.56±.10 | 0.55±.10 | 0.48±.07 |
| $LSTM^*$ | 0.98±.04 | 0.95±.03 | 0.88±.05 | 0.87±.04 | 0.81±.07 | 0.75±.10 | 0.75±.09 |
| $GRU^*$ | 0.89±.05 | 0.83±.06 | 0.74±.12 | 0.72±.09 | 0.67±.12 | 0.62±.10 | 0.60±.12 |

## G.2 EXTENDED ABLATION ANALYSIS

In the main paper, we considered four separate ablations. Table 14 extends this analysis by showing results for all combinations of these ablations. Facet ablation refers to the configurations where $m = 1$; probability ablation refers to the configuration where embeddings are unconstrained; composition ablation refers to the configuration where distmult in combination with an MLP is used as the composition function $\psi$; and backward ablation refers to the configuration where we only have the forward model. We can clearly see that the probability and composition function ablation cause a significantly stronger performance degradation compared to the facet and forward-backward ablation.

Table 14: Results for all combinations of the individual ablations from Table 4.

| Facet Ablation | Probability Ablation | Composition Ablation | Backward Ablation | CLUTRR Avg | CLUTRR $k = 10$ | RCC-8 Avg | RCC-8 $b = 3, k = 9$ |
|---|---|---|---|---|---|---|---|
| True | True | True | True | 0.06 | 0.04 | 0.12 | 0.12 |
| True | True | True | False | 0.10 | 0.15 | 0.12 | 0.12 |
| True | True | False | True | 0.27 | 0.24 | 0.25 | 0.17 |
| True | True | False | False | 0.20 | 0.16 | 0.25 | 0.14 |
| True | False | True | True | 0.06 | 0.04 | 0.12 | 0.12 |
| True | False | True | False | 0.11 | 0.20 | 0.12 | 0.12 |
| True | False | False | True | 0.92 | 0.73 | 0.81 | 0.49 |
| True | False | False | False | 0.94 | 0.85 | 0.92 | 0.68 |
| False | True | True | True | 0.06 | 0.04 | 0.22 | 0.18 |
| False | True | True | False | 0.11 | 0.15 | 0.12 | 0.12 |
| False | True | False | True | 0.29 | 0.25 | 0.60 | 0.27 |
| False | True | False | False | 0.36 | 0.30 | 0.38 | 0.21 |
| False | False | True | True | 0.08 | 0.10 | 0.12 | 0.12 |
| False | False | True | False | 0.29 | 0.31 | 0.12 | 0.12 |
| False | False | False | True | 0.94 | 0.82 | 0.84 | 0.51 |
| False | False | False | False | 0.99 | 0.99 | 0.96 | 0.80 |

## G.3 LEARNED SPARSENESS OF RELATION VECTORS

We visualize the learned relation vectors for each benchmark studied in this paper in Figures 9, 10, 11 and 12. It can be seen that the vectors are mostly one-hot, despite the fact that no explicit sparsity constraints were used in the model. Also, we note that different facets are capturing different parts of a relation and there is shared structure between different relations if the semantic meaning is similar

e.g. contrast the vector for grandfather and grandmother or husband and wife in Figure 11, and similarly, the vectors for si, s share a structure each being the other's inverse in Figure 10.

Note that for the relation prediction task, there are no entities at each node but rather intermediate compositions of relations.

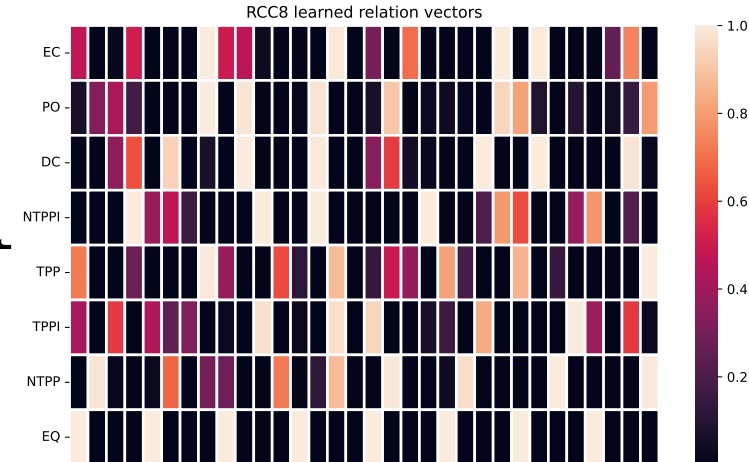

Figure 9: Schematic visualisation of the learned relation vectors with 4 facets with a hidden dimension of 8 for the RCC-8 benchmark. Notice that the eq relation is learned to be one-hot at the first index for every facet as it corresponds to the identiy composition in Eq. (4).

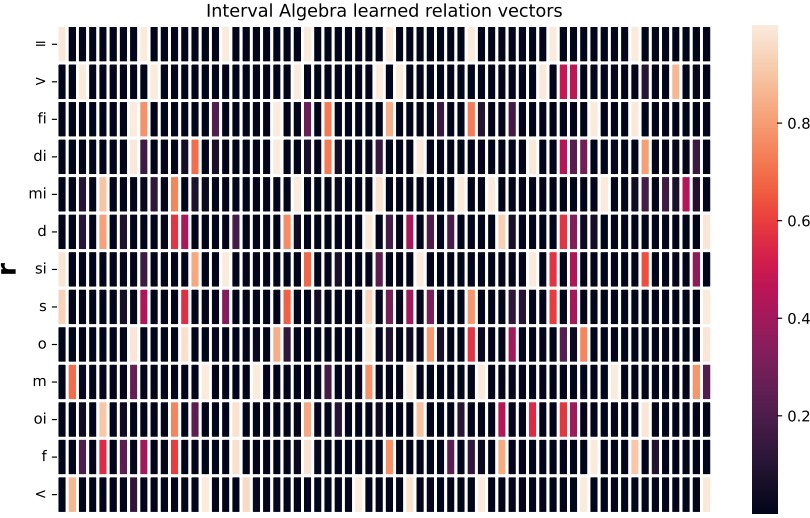

Figure 10: Schematic visualisation of the learned relation vectors for the Interval Algebra benchmark with 8 facets, each with a hidden dimension of 8. Again, the learned identity relation representation = corresponds to the identity composition.

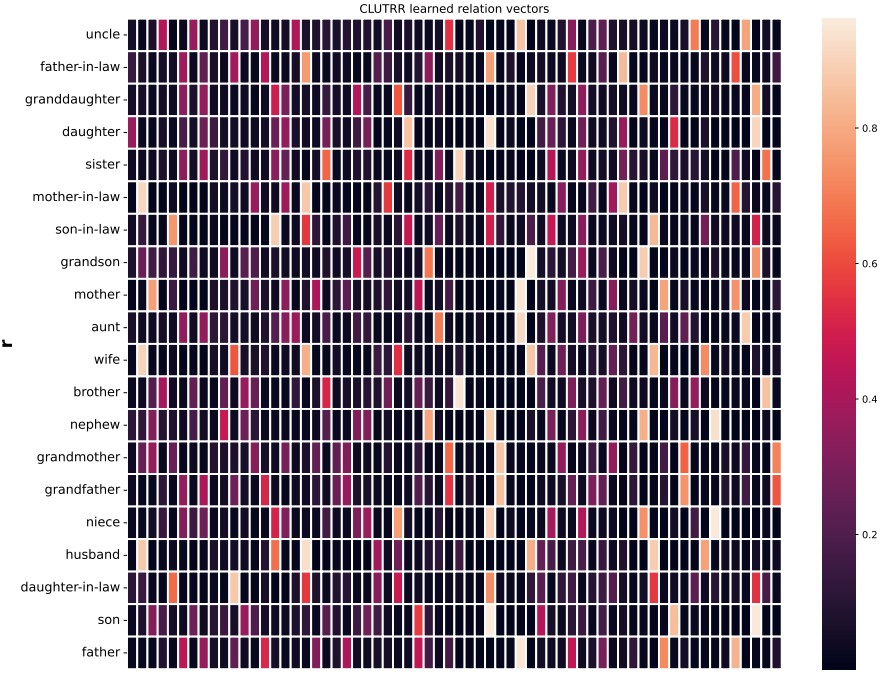

Figure 11: Schematic visualisation of the learned relation vectors for the CLUTRR benchmark.

## G.4 EFFECT OF THE NUMBER OF MESSAGE PASSING ROUNDS

We study the effect of varying the number of message passing rounds on the accuracy for varying values of $k$ and $b = 3$, for the EpiGNN-`min` and EpiGNN-`mul` models on the RCC-8 and IA datasets. These models are trained *ab initio* with the same training data and configuration as before but with a different number of message passing rounds (from 5 to 15) for each instance. The results are displayed in Figure 13. There are three pertinent observations that can be made. Firstly, the maximum attained $k$-hop accuracy decreases with $k$, which makes sense as it confirms that the problem complexity increases with $k$. Secondly, there is a jump in the $k$-hop accuracy when the number of message passing rounds matches $k$ after which the accuracy saturates. This rightly suggests that the number of message passing rounds should at least be equal to the final $k$-hop in the dataset to ensure that all information propagates from head entity to tail within the model. Thirdly, these observations are shared across the dataset types and aggregation functions.

## G.5 ADDITIONAL ANALYSIS OF PARAMETER AND TIME COMPLEXITY

**Knowledge graph completion** The empirical parameter complexity of the EpiGNN on all splits (Teru et al., 2020) of the inductive knowledge graph completion benchmarks for FB15k-237 (Toutanova & Chen, 2015) and WN18RR (Dettmers et al., 2018b) is shown in Figure 14. The parameter estimates of the best performing GNN baselines in Table 3, namely NBFNet (Zhu et al., 2021) and REST (Liu et al., 2023) are also shown. It can be observed that the EpiGNN is the most parameter efficient model out of all 3 by at least on order of magnitude on all the splits. The time complexity with respect to NBFNet is shown in Figure 15 where the EpiGNN is slightly slower than NBFNet on FB15k-237 for both training and inference but faster for both in WN18RR.

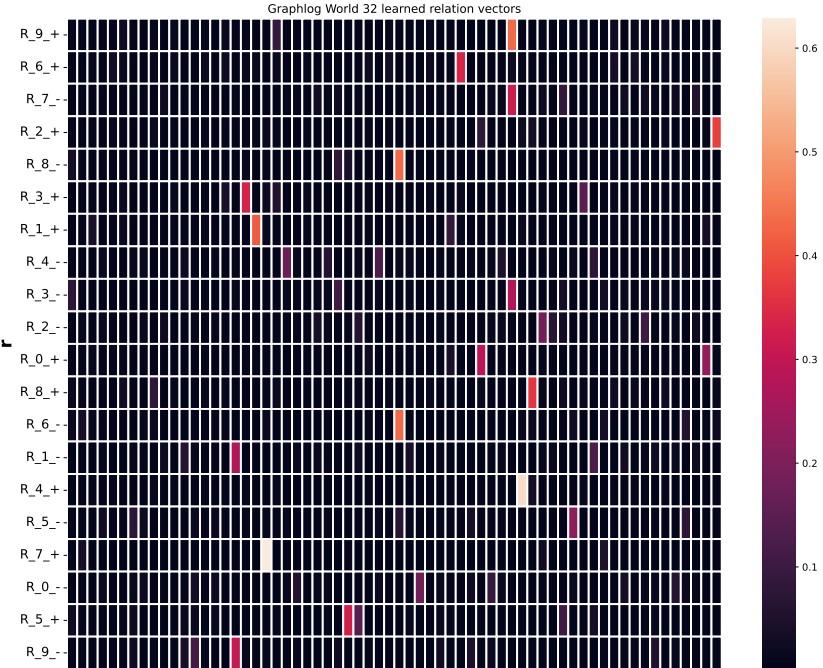

Figure 12: Schematic visualisation of the learned relation vectors for the Graphlog benchmark.

**Spatio-temporal reasoning** The training and inference times of the EpiGNN with respect to the edge transformers (Bergen et al., 2021) (cf. the best baseline in Figure 4) are shown in Figure 16.

### G.6 RESULTS ON THE EXPANDED VERSIONS OF RCC-8 AND INTERVAL ALGEBRA DATASETS

The RCC-8 and Inverval algebra datasets presented in the main text can be made more challenging by increasing the number of paths $b$ and the maximum number of inference hops $k$. In this section, we provide results for the expanded RCC-8 and IA datasets for the EpiGNN and Edge Transformers (being the best baseline in Figure 4).

The Edge transformer is not able to fit graphs in memory on an RTX 4090 GPU with $k > 9$ and $b \geq 6$ so we need to restrict the comparison in Figure 17. The EpiGNN is able to handle much larger graphs since it is more compute and parameter efficient so we provide the results for up to $k = 15$ and $b = 8$ in Figure 18. Note that our model holds its performance fairly steady on this significantly expanded dataset with an average accuracy of 0.74 on RCC-8 and 0.77 on the IA dataset for the hardest setting: $k = 15$ and $b = 8$. Edge Transformers significantly deteriorate on IA and can only achieve an average accuracy of 0.19 at $k = 9, b = 6$. On RCC-8, the Edge transformer has an average accuracy of 0.7 for $k = 9, b = 6$ versus 0.85 for our model.

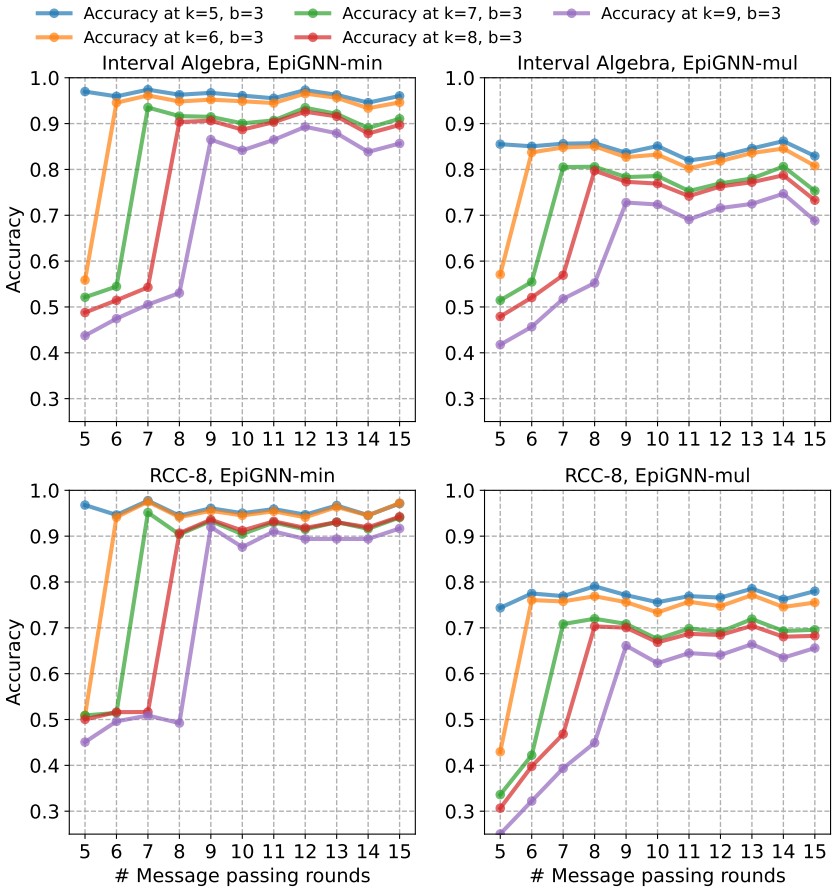

Figure 13: Effect of the number of message passing rounds on the $k$-hop accuracy for the EpiGNN model on the IA and RCC-8 datasets. There is a sharp jump in performance when $k$ equals the number of message passing rounds.

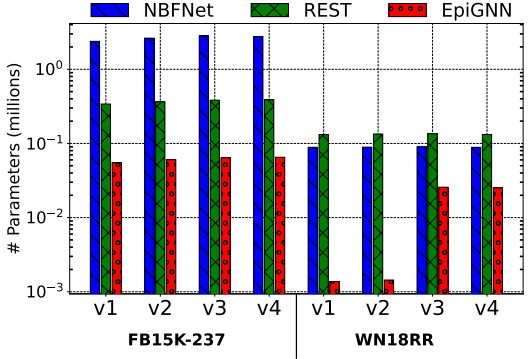

Figure 14: Parameter complexity on all the inductive versions of FB15k-237 and WN18RR.

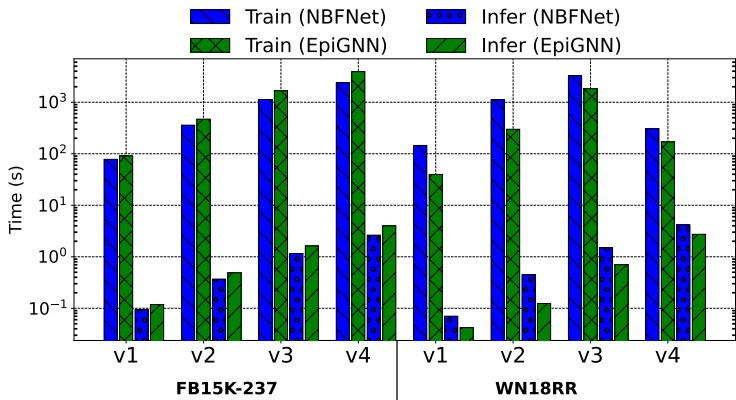

Figure 15: Time complexity of the EpiGNN against the NBFNet on all the inductive versions of FB15k-237 and WN18RR. Results are obtained on a single Nvidia RTX 4090 GPU.

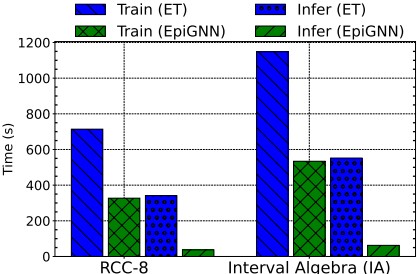

Figure 16: Time complexity of the EpiGNN against the best baseline on spatio-temporal systematic reasoning. Results are obtained on a single Nvidia RTX 4090 GPU.

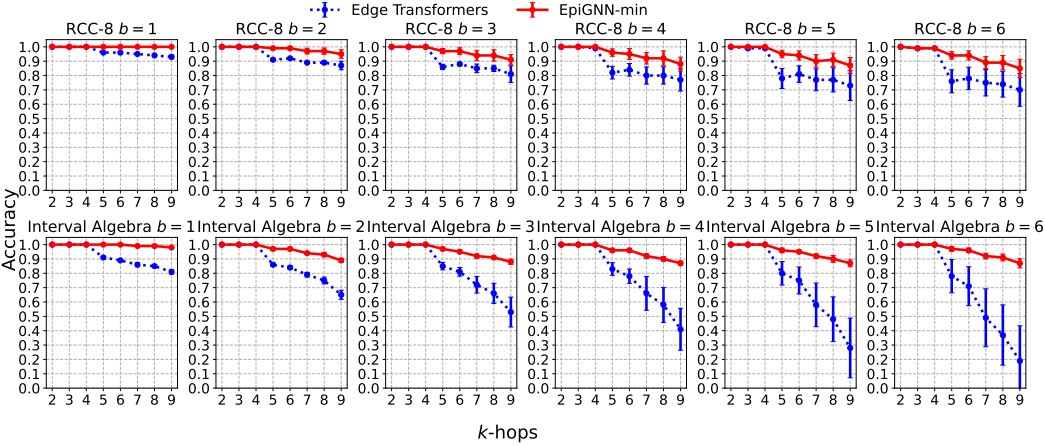

Figure 17: Performance of EpiGNN-`min` and Edge Transformers on the expanded version of the RCC-8 and Interval Algebra datasets. The performance of Edge Transformers deteriorates as the number of paths is increased compared to EpiGNNs and this effect can be more significantly observed on the Interval Algebra dataset. We restrict the comparison to $b \leq 6, k \leq 9$ since Edge transformers cannot fit graphs for the other settings in memory on an RTX 4090 GPU.

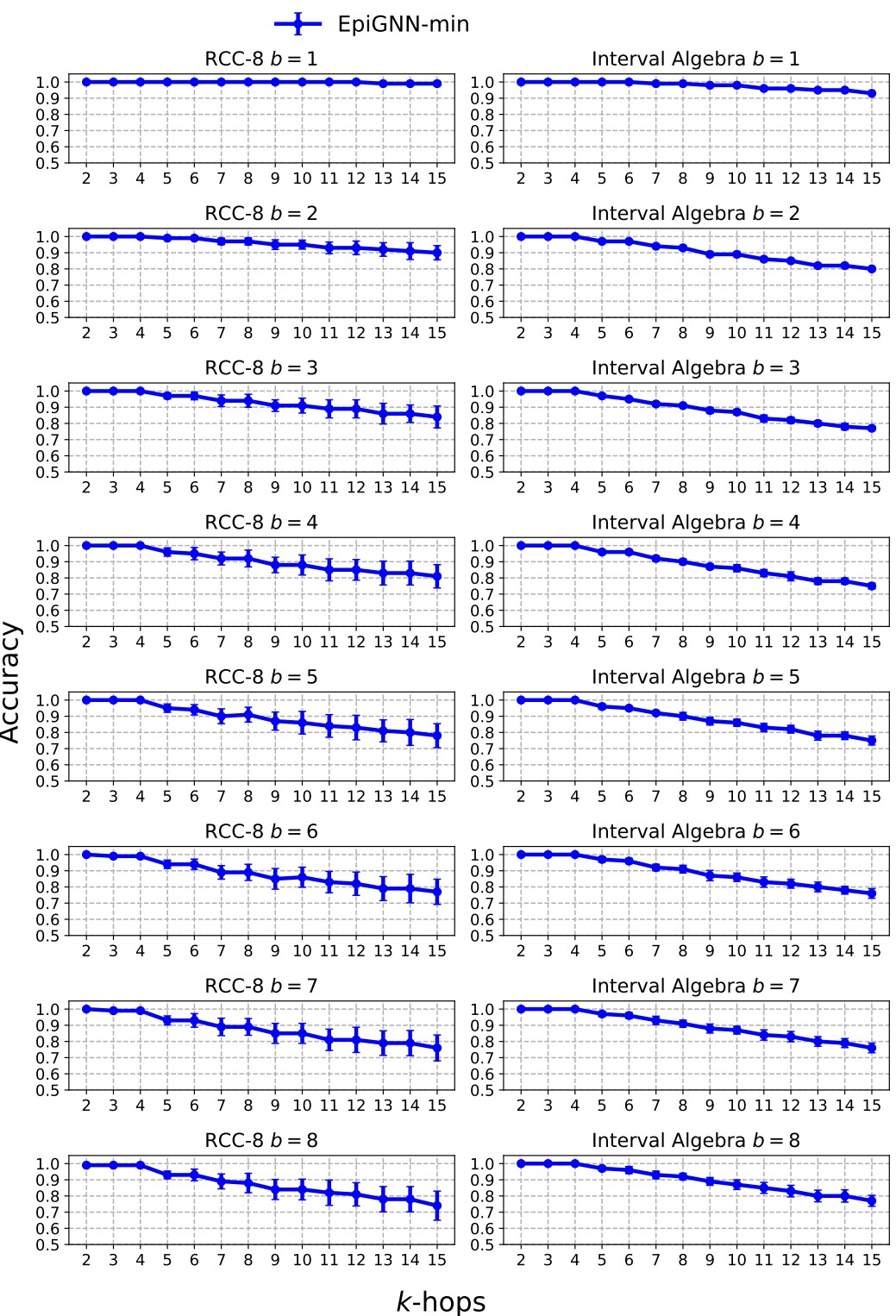

Figure 18: Performance of EpiGNN-min on the complete expanded version of the RCC-8 and Interval Algebra datasets with maximum values of $b = 8, k = 15$. The model's performance is scalable and is fairly steady for the hardest setting: $b = 8, k = 15$ highlighting its inductive bias for systematic generalization.

