# OpenReview forum: "Systematic Relational Reasoning With Epistemic Graph Neural Networks"
_ICLR.cc/2025/Conference — ICLR 2025 Poster_

### Official Review · Reviewer_tM14 · 2024-11-01

**Soundness:** 3
**Presentation:** 3
**Contribution:** 2
**Rating:** 8
**Confidence:** 4

**Summary:**

The paper studies systematic generalization in the domain of multi-relational graphs. First, the authors identified that existing neuro-symbolic baselines capture only conjunctive rules and cannot capture disjunctive rules often present in spatial and temporal reasoning settings (such as Region Connected Calculus or Interval Algebra). Then, the paper introduced EpistemicGNN, which is able to learn disjunctive rules thanks to the theoretical alignment with algebraic closure. Practically, EpistemicGNN extends Neural Bellman-Ford Nets (NBFNet) with a new node labeling strategy, constrained node representations, changes to message passing and aggregation. For experiments, the authors introduced two new datasets for relation classification (RCC-8 and IA) and showed that EpiGNN is competitive with neuro-symbolic methods while being more parameter-efficient.

**Strengths:**

**S1.** Theoretical findings are quite interesting: a link between path-based GNNs and algebraic closure and how to learn disjunctive rules with such GNNs. Since EpiGNN is based on NBFNets, it would make the theoretical expressiveness claims even stronger if there is a connection to the relational WL test [1] which was derived for simple relational GNNs (eg, R-GCN) and labeling trick GNNs (like NBFNet).

**S2.** Two new benchmarks for measuring systematic generalization with disjunctive reasoning patterns are nice to have for the community. However, there is a caveat that those benchmarks seem to be already saturated given that EpiGNN yields 90+% accuracy on them.

[1] Huang et al. A Theory of Link Prediction via Relational Weisfeiler-Leman on Knowledge Graphs. NeurIPS 2023.

**Weaknesses:**

**W1.** Despite the variety of benchmarks, the experimental performance of EpiGNNs is somewhat underwhelming: CLUTTR is already a saturated benchmark that does not provide much useful signal, Edge Transformers are as good on two newly introduced benchmarks (which seem to be saturating already), and EpiGNN underperforms other baselines on GraphLOG and inductive KG completion benchmarks.

It would make KG completion experiments much more compelling if EpiGNN is evaluated in the setup when predictions are ranked against _all_ nodes in the graph, not just against 50 random samples (there are many studies indicating that such ranking overestimates the results and is not indicative of the real performance). Since the backbone of EpiGNN is NBFNet, it should be possible to rank against all entities right out of the box without modifications and compare against stronger baselines that were evaluated on the full ranking task (for instance, NBFNet, A*Net, RED-GNN, AdaProp).

**Questions:**

I’d be willing to increase the score if the authors could address the following questions:

Q1. Is there a link to the Relational WL test [1] when estimating the expressiveness of EpiGNNs?

Q2. KG completion evals on GraIL datasets should be executed in the full-ranking setup, not against 50 random negatives.

Q3. The new benchmarks should be future-proof for a few years ahead whereas RCC-8 and IA already seem to be saturating with models delivering 80%+ accuracy on hardest cases. Would it be possible to design more complex versions of the datasets where all compared models including EpiGNN would be challenged?

---

### Official Review · Reviewer_JqVR · 2024-11-04

**Soundness:** 3
**Presentation:** 4
**Contribution:** 2
**Rating:** 6
**Confidence:** 5

**Summary:**

This paper aims to solve systematic generalization in relational reasoning problems. Existing GNNs lack the ability to generalize systematically, while existing neuro-symbolic methods can generalize but tend to make strong assumptions. The authors propose Epistemic GNNs (EpiGNNs), which not only solves systematic generalization but is also scalable and parameter efficient. Technically, EpiGNNs modify the NBFNet architecture to represent representations between entities as distributions over primitive relationships. The messages are then defined as learnable weighted compositions of these primitive relations. Besides, the authors also propose a few tricks to augment EpiGNNs, including jointly training several models and two forward & backward models. The authors introduce two new datasets for benchmarking the ability of handling disjunctive relational paths. Empirically, EpiGNNs surpass all methods on the two new datasets. EpiGNNs also achieve competitive performance against existing methods on systematic generalization and inductive knowledge graph completion datasets.

**Strengths:**

- This paper introduce two new datasets for benchmarking systematic generalization. The two new datasets consider multi-path disjunctive rules, which is challenging for most existing methods.
- EpiGNNs achieve state-of-the-art performance on the two new datasets. EpiGNNs also achieve competitive performance against existing methods on systematic generalization datasets, while being more efficient.
- EpiGNNs have theoretical connection with the algebraic closure algorithm, though this point is not illustrated much in the main paper.
- This paper is well-written and easy to comprehend.

**Weaknesses:**

- EpiGNNs are variants of NBFNet[1] with some engineering modifications: 1) hidden representations are replaced by probability distributions; 2) DistMult message function is replaced by Tucker decomposition[2] (Equation 4); 3) pooling function has an additional L1 normalization; 4) joint training of multiple models and forward-backward models. While I am not saying that EpiGNNs are incremental, the authors need to show how these modifications contribute to systematic generalization and provide experimental evidence for Line 257-258.
- The authors miss an important baseline for systematic generalization: EdgeTransformer[3]. EdgeTransformer aggregates multiple relational paths like EpiGNNs and NBFNet. It can solve the sequential problem (Line 146-148) in arbitrary reduction order, which is even better than EpiGNNs. EdgeTransformer is likely to be a strong baseline for the two new datasets.
- If I understand correctly, EpiGNNs use the same parameter matrix $\bf{a}_{ij}$ across all layers. The original NBFNet uses different parameters for different layers and is not ideal for systematic generalization. Can you try a stronger NBFNet baseline with parameters shared across all layers? You might need to add dropout in each layer of NBFNet, similar to the design of EdgeTransformer.
- It is trivial for EpiGNNs to achieve competitive performance against NBFNet on inductive knowledge graph completion, since EpiGNNs are just variants of NBFNet. This is not a contribution.

[1] Neural Bellman-Ford Networks: A General Graph Neural Network Framework for Link Prediction. Zhu et al. NeurIPS 2021.

[2] TuckER: Tensor Factorization for Knowledge Graph Completion. Balažević et al. arXiv 2019.

[3] Systematic Generalization with Edge Transformers. Bergen et al. NeurIPS 2021.

**Questions:**

- Line 50 & 53: It is more precise to say neural theorem provers rather than neuro-symbolic methods. Neuro-symbolic methods are a very broad family, and not all of them focus on modeling single relational paths.
- Proposition 1: I guess there exist scenarios where the count of a certain relational path matters, right? Maybe you can take this into consideration.
- Line 154-155: This statement is not true for representation learning methods like NBFNet and EpiGNNs. If the reduction of relations follows the associative property (i.e. $(a\times b)\times c=a\times (b\times c)$), representation learning methods can deal with intermediate relations that are not present in the training data.
- Can you include a few references for epistemic states if this concept is not invented by this paper?
- Line 269-271: The pairwise disjoint assumption only holds for CLUTRR and Graphlog. It doesn’t hold for knowledge graph datasets in general.
- Typo: Line 100: single-path base → single-path-based

---

### Official Review · Reviewer_C39j · 2024-11-04

**Soundness:** 2
**Presentation:** 3
**Contribution:** 3
**Rating:** 6
**Confidence:** 3

**Summary:**

This paper focuses on a challenge on current knowledge graph inference study that learned model fails to generalize to test cases with longer logical inference path. This paper modifies the state-of-the-art NBFNet design, and proposes EpiGNN, a scalable model which is proved to be same expressive as multi-path disjunctive reasoning that is enough to solve the relation inference of any length. They proposes two benchmarks CLUTRR and GraphLog to evaluate longer relational path inference, and empirically showcase that their method can achieve same good performance as state-of-the-art methods but with better efficiency. They also empirically shows that their model can achieve close-to-best performance on real-world benchmarks where longer logic inference may not be necessary.

**Strengths:**

- This paper focuses on longer logic inference with requirement of multi-path disjunctive reasoning, which are critical problems in knowledge graph.
- This paper's solution is based on scalable framework, making it useful in giant knowledge graphs in real world.
- This paper has prove of expressivity in the sense of logical reasoning, promising its power to handle focusing problems.

**Weaknesses:**

- Some design details have potentially alternatives, but is lack of reasoning, leaving concerns that if the implementation is the best design under proposed framework.
- The empirical evidence is not strong enough to showcase their claims:
   - The studying problem is claimed to be unsolvable for existing methods, while baselines can achieve the best performance.
   - The method is claimed to be more scalable than related baselines, however, there is runtime or memory cost comparison in main content.
   - In real-world benchmarks, their method is not such close to state-of-the-art as they claimed. I think adding confidence interval will better defend their point here.
- Their design does not significantly differ from related work NBFNet, but is lack of theoretical justification why their method can be more powerful than this related work.

**Questions:**

1. line 113: Shouldn't $\mathcal{K} \circ \mathcal{F}$ (applies on) better than $\mathcal{K} \cup \mathcal{F}$ (union). Since $\mathcal{K}$ are rule set like $r_1 \wedge r_2 = r_3$ while $\mathcal{F}$ are facts like $r_1(a, b), r_1(b, c), r_2(c, d)$?
2. Equation 3: Based on your reasoning, this initialization hints the probability of some primitive relations. Then, it seems that the first one is if or not it is the anchor node (zero-one init as NBFNet). If that undertstanding is correct, shouldn't the second be $(0, 1/(n-1), \dots)$?
3. line 236: This equation is slightly different from tradition GNN update rule in the sense that node embedding itself is considered as a message (self-loop edge), and update function is removed, any hint behind doing that?
4. line 244: It will be better to use $\mathbf{A}$ instread of $\emph{\mathbf{a}}$ since it is a 3D tensor rather than a vector.
5. line 269: Is the relation being pairwise disjoint is the assumption only on your dataset? I don't think this is true for most real-world knowledge graphs.
6. line 275-280: If we add multi facet only at loss function, we are training them independently, shouldn't be better to add a pooling at final output of each layer, just as multi-head attention?
7. Equation 6: You are only merging bidirections on both ends, but won't it be possible that we have to merge relations in the middle first? If so, shouldn't we merge at the output of every layer?
8. An overall concern: Your design eventually is the same as message passing on query anchored graph (except the final bidrection aggregation), which is proposed in NBFNet (except initialization and message passing function). My question is which unique design makes it more capable than NBFNet, thus can solve problem it can not handle?
9. Table 1: Why NBFNet is worse than GCN, GAT and RGCN in many cases? Does that mean that NBFNet is extremely poor in true logic reasoning? One more thing, it seems that you are only picking samples that relations must exist which is slightly different from NBFNet purpose for link prediction. Can you also add unrelated samples in the test? Indeed, in Table 3 where such negative samples exist, your method is not performing such better, thus this may mean your method fails to handle unrelated node pairs which I believe could be a big risk in real-world applications.
10. Table 1 and 2: It is claimed before than NCRL and R5, as single-path method, is not capable to solve the studying problem (line 183), however, they are mostly the best on two proposed benchmarks, doesn't that means your benchmarks are not the best fit for evaluation? Shouldn't you additionally propose a synthetic benchmark, that truly justify your method can handle multi-path reasoning?
11. Based on overall experiment result, the advantage of your proposal is more efficient in longer relation path generation, thus can you provide the real runtime (training and inference) comparison with NCRL and R5 (which achieves best performance on same scenarios).

---

### Official Review · Reviewer_X3MA · 2024-11-05

**Soundness:** 3
**Presentation:** 3
**Contribution:** 2
**Rating:** 6
**Confidence:** 2

**Summary:**

This work presents a representation-based method for predicting relationships between entities within a knowledge graph. The approach leverages Graph Neural Networks (GNNs) and incorporates epistemic states as node representations. The proposed method is benchmarked against various rule-based and graph-based approaches, demonstrating superior performance.

**Strengths:**

1. The method introduced here demonstrates significantly higher efficiency compared to other baselines, as clearly illustrated in Figure 5.
2. The proposed method is compared against various baselines, including both rule-based and graph-based approaches, and consistently shows superior performance.
3. New synthetic datasets are provided in this work.

**Weaknesses:**

1. In my opinion, representing each node/entity as a distribution over relations is unconventional. Despite its parameter efficiency and superior performance, this embedding/representation method has notable drawbacks: (1) The learned embeddings are non-transferable across datasets, as different datasets may have varying numbers of relations. (2) It completely ignores the semantic information of entities, making the semantic interpretation of these representations difficult.
2. The notations in Section 3 are somewhat challenging to follow. For instance, what does 𝑓 represent in Eq. 4 and the equation preceding it? Additionally, why is 𝑓 not illustrated in Figure 3? Furthermore, in Eq. 4, how is a_ij ​ defined? Providing an example of this variable or illustrating it within the message passing subplot in Figure 3 could enhance interpretability.
3. I am not an expert in this field, but I wonder if this method can be applied to entity prediction. From my perspective, entity prediction is more interesting and challenging in knowledge graphs (KG) and temporal knowledge graphs (tKG) [1, 2]. It seems that relation prediction is relatively simpler, as the number of potential relations to predict is usually limited to a few hundred. Can the method proposed in this work be generalized to entity prediction, or is it restricted to relation prediction?

[1] Anytime Bottom-Up Rule Learning for Knowledge Graph Completion
[2] TLogic: Temporal Logical Rules for Explainable Link Forecasting on Temporal Knowledge Graphs

**Questions:**

See above weakness

---

### Meta-Review · Area_Chair_K4Rj · 2024-12-17

**Metareview:**

This paper introduces Epistemic GNN (EpiGNN), a novel GNN architecture designed for systematic reasoning in relational domains. EpiGNN overcomes the limitations of regular GNNs by treating node embeddings as epistemic states and using specialized message passing. It achieves state-of-the-art results in link prediction and knowledge graph completion tasks, and outperforms existing methods in new benchmarks that require aggregating information from multiple paths.

This paper is well-written and easy to comprehend. Two new benchmarks for measuring systematic generalization with disjunctive reasoning patterns are nice to have for the community.

**Additional Comments On Reviewer Discussion:**

All reviewers agree with an acceptance.

---

### Decision · Program_Chairs · 2025-01-22

Accept (Poster)